# ARM-FM: Automated Reward Machines via Foundation Models for Compositional Reinforcement Learning

**Roger Creus Castanyer**[1,2]     **Faisal Mohamed**[1,2]     **Pablo Samuel Castro**[1,2]

**Cyrus Neary**[3]                                            **Glen Berseth**[1,2]

[1] Université de Montréal     [2] Mila – Quebec AI Institute     [3] University of British Columbia

## Abstract

Reinforcement learning (RL) algorithms are highly sensitive to reward function specification, which remains a central challenge limiting their broad applicability. We present ARM-FM: Automated Reward Machines via Foundation Models, a framework for automated, compositional reward design in RL that leverages the high-level reasoning capabilities of foundation models (FMs). Reward machines (RMs) – an automata-based formalism for reward specification – are used as the mechanism for RL objective specification, and are automatically constructed via the use of FMs. The structured formalism of RMs yields effective task decompositions, while the use of FMs enables objective specifications in natural language. Concretely, we (i) use FMs to automatically generate RMs from natural language specifications; (ii) associate language embeddings with each RM automata-state to enable generalization across tasks; and (iii) provide empirical evidence of ARM-FM's effectiveness in a diverse suite of challenging environments, including evidence of zero-shot generalization.

## 1 Introduction

A central challenge in reinforcement learning (RL) is the design of effective reward functions for complex tasks. The *shape* of the reward influences the complexity of the problem at hand (Gupta et al., 2022); for instance, sparse rewards provide an insufficient learning signal, making it difficult for agents to improve (Devidze et al., 2022). Even hand-crafted dense rewards are susceptible to unintended loopholes or "reward hacking", where an agent exploits the specification without achieving the true objective (Fu et al., 2025). The unifying challenge is thus how to communicate complex objectives to an agent in a manner that provides structured, actionable guidance (Rani et al., 2025).

While Foundation Models (FMs) excel at interpreting and decomposing tasks from natural language, a critical gap exists in translating this abstract understanding into the concrete structured reward signals necessary for RL. Consequently, high-level plans generated by FMs often fail to ground effectively, leaving the agent without the granular feedback required for learning. To bridge this gap, we turn to Reward Machines (RMs), an automata-based formalism. By decomposing tasks into a finite automaton of sub-goals, RMs provide a compositional structure for both rewards and policies that is inherently more structured and verifiable than monolithic reward functions (Icarte et al., 2022). While theoretically principled, their practical application has been confined to task-specific applications due to the complexity of their manual, expert-driven design. We posit that the reasoning and code-generation capabilities of modern FMs are well-suited to automate the design and construction of RMs, thereby unlocking their potential to solve the broader challenge of communicating complex objectives in RL; the resulting RMs can thus translate abstract human intent into a concrete learning signal for solving complex tasks.

This work makes three primary contributions. First, we develop a novel framework for automatically generating complete task specifications directly from natural language using foundation models,

introducing language-aligned reward machines (LARMs) which include the automaton structure, executable labeling functions, and natural language instructions for each subtask. Second, we introduce a method that leverages the language-aligned nature of the resulting automata to create a shared skill space, enabling effective experience reuse and policy transfer across related tasks. Finally, we provide extensive empirical validation demonstrating that our approach solves complex, long-horizon tasks across multiple domains that are generally intractable for standard RL methods. Specifically, our results show that the framework (i) dramatically improves sample-efficiency by converting sparse rewards into dense, structured learning signals; (ii) scales to a diverse set of environments, including grid worlds, complex 3D environments, and robotics with continuous control; and (iii) enables efficient multi-task training and zero-shot generalization.

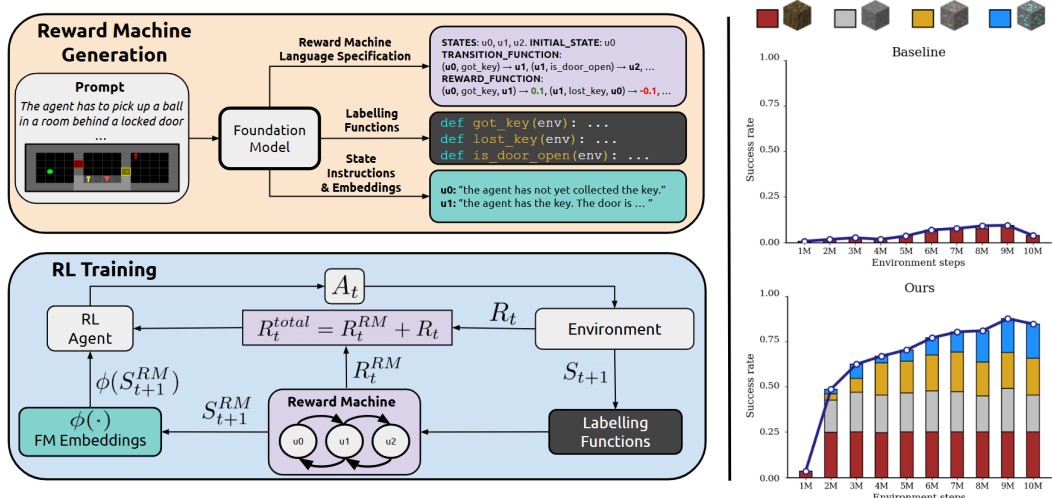

Figure 1: An overview of our framework (left) and results in a complex sparse-reward environment (right). **Reward Machine Generation (top-left):** Given a high-level natural language prompt and a visual observation of the environment, a FM automatically generates the formal specification of the Reward Machine, the executable Python code for the **labeling functions**, and the natural language descriptions for each RM state. **RL training (bottom-left):** During the RL training loop, the **labeling functions** evaluate environment observations to update the Reward Machine's state, which provides a dense reward signal $R_t^{\mathrm{RM}}$. The RL agent's policy receives the environment observation along with the **embedding** $\phi(\cdot)$ of the current RM state's language description, making it aware of its active sub-goal. **Empirical results (right):** Results in a complex sparse-reward Minecraft-based resource-gathering task from `Craftium` (Malagón et al., 2024), where an RL agent is unable to make progress (top), while our agent, guided by an FM-generated LARM, learns to solve the task efficiently (bottom).

## 2   AUTOMATED REWARD MACHINES VIA FOUNDATION MODELS

We now present **Automated Reward Machines via Foundation Models** (ARM-FM), a framework for automated reward design in RL that leverages the reasoning capabilities of foundation models to automatically translate complex, natural-language task descriptions into structured task representations for RL training. Figure 1 illustrates an overview of ARM-FM, which comprises two major components: *(i)* the introduction of **Language-Aligned RMs** (LARMs), which are automatically constructed using FMs; and *(ii)* their integration into RL training by conditioning policies on language embeddings of RM states, enabling structured rewards, generalization, and skill reuse. Figure 2 shows a high-level task description, consisting of a natural language prompt and a visual observation (left), along with its corresponding RM (right). The RM is a finite-state automaton that guides an agent by providing incremental rewards for completing sub-goals, such as collecting keys and opening doors, on the way to the final objective. In the following section, we describe how our ARM-FM framework automates the creation of these reward machines directly from high-level task descriptions.

## 2.1 LANGUAGE-ALIGNED REWARD MACHINES

We assume the standard RL formalism, which defines an environment as a Markov Decision Processes (Puterman, 1994, ; MDPs) $\langle S, A, \mathcal{R}, \mathcal{P} \rangle$, where $S$ is the set of MDP states, $A$ is the set of possible actions, $\mathcal{R} : S \times A \to \mathbb{R}$ is the MDP reward function, and $\mathcal{P} : S \times A \to \Delta(S)$ is a probabilistic MDP transition function. A **Reward Machine** (RM) is a finite-state automaton that encodes complex, temporally extended, and potentially non-Markovian RL tasks (Icarte et al., 2022). We formally define an RM by the tuple $\langle U, u_I, \Sigma, \delta, R, F, \mathcal{L} \rangle$. Here, $U$ is the finite set of RM states; $u_I$ the initial state

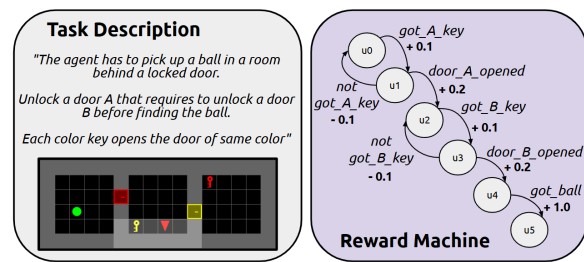

Figure 2: **ARM-FM leverage FMs to automatically construct RMs:** using the `UnlockToUnlock` task description from `MiniGrid` (left), an RM is automatically constructed to solve the task (right).

of the RM; $\Sigma$ is the finite set of symbols representing events that cause transitions in the RM; $\delta : U \times \Sigma \to U$ is the deterministic RM transition function; $R : U \times S \times A \times S \to \mathbb{R}$ is the RM reward function; $F \subseteq U$ is the set of final RM states; and $\mathcal{L} : S \times A \to \Sigma$ is the labeling function that connects MDP states $s \in S$ and actions $a \in A$ to the RM event symbols $\sigma \in \Sigma$. Intuitively, RMs are useful for describing tasks at an abstract level, especially when said tasks require multiple steps over long time horizons. Each RM state $u \in U$ can be thought of as representing a subtask, and the transitions $u' = \delta(u, \sigma)$ denote progress to a new stage of the overall objective after a particular event $\sigma \in \Sigma$ occurs in the environment. The RM reward function $R(u, s, a, s')$ assigns a reward based on the current RM state $u$ and the underlying MDP transition $(s, a, s')$. Meanwhile, the set of final RM states $F$ defines the conditions under which the task described by the RM is complete. Finally, the labeling function $\mathcal{L}$ is required to connect the RM's events, transitions, and rewards to states and actions from the underlying MDP.

We define LARMs as RMs that are additionally equipped with natural-language instructions $l_u$ for each RM state $u$, and with an embedding function $\phi(\cdot)$ that maps such language instructions to an embedding vector $z_u = \phi(l_u) \in \mathbb{R}^d$. We note that by equipping RM states with embedding vectors $z_u$ that encode language-based descriptions of the corresponding subtasks, we provide the first mechanism for constructing a *semantically grounded skill space* in RMs: policies conditioned on these embeddings can naturally share knowledge across related subtasks, enabling transfer, compositionality, and zero-shot generalization.

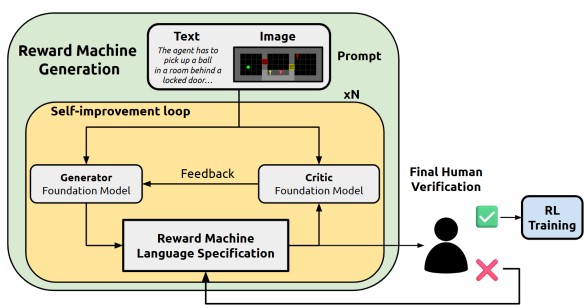

Figure 3: A self-improvement loop where a generator and critic FMs iteratively refine LARMs, with optional human verification.

We present a framework to automatically construct LARMs from language-and-image-based task descriptions by iteratively prompting an FM, as is illustrated in Figure 3. More specifically, to progressively refine the RM specification, we employ $N$ rounds of self-improvement using paired *generator* and *critic* FMs (Tian et al., 2024). A human may optionally intervene by approving the output or providing corrective feedback (see Appendix A.4 for details). In practice, we find that FM-generated reward machines are both interpretable and easily modifiable, as they follow a natural language specification. Figure 4 illustrates an automatically-constructed LARM for the `UnlockToUnlock` task, including a text-based description of the RM (left), FM-generated labeling functions $\mathcal{L}$ (middle), and natural RM state instructions and embeddings $l_u$ (right). All RMs and labeling functions used in this work are shown in Appendix A.9 and A.10. While we use code to define labeling functions in this work, the ARM-FM framework is general, supporting any boolean predicate (e.g. formal logic, or queries to other FMs).

## 2.2 Reinforcement Learning with LARMs

The introduction of the LARM uses an augmented state space that is the cross-product of the MDP and RM states ($\mathcal{S} \times \mathcal{U}$), and a reward function that is the sum of the MDP and RM rewards; we will refer to this augmented MDP as $\mathcal{M}'$, and it is illustrated in Figure 1 (Bottom). At timestep $t$, the agent selects actions conditioned on the environment state and the language embedding of the current LARM state: $\pi(s_t, z_{u_t})$. This language-based policy conditioning is the central mechanism enabling generalization in our framework, creating a semantically grounded skill space where instructions like "pick up a blue key" and "pick up a red key" are naturally close in the embedding space, unlocking a pathway for broad experience reuse and efficient policy transfer.

During training, after the agent executes an action $a_t \sim \pi(s_t, z_{u_t})$, the underlying MDP transitions to $s_{t+1}$ and returns a reward $R_t$. The labeling function $\mathcal{L}(s_{t+1}, a_t)$ determines if a symbolic event has occurred, which may induce a LARM transition $u_{t+1} = \delta(u_t, \mathcal{L}(s_{t+1}, a_t))$, as well as an additional reward $R_t^{\text{RM}}$. The sum of the MDP and RM rewards are then used for learning: $R_t^{\text{total}} = R_t + R_t^{\text{RM}}$. This complete training procedure, adapted for a DQN agent, is formalized in Appendix A.3. The effectiveness of well-designed LARMs yields theoretical guarantees (see Appendix A.5), ensuring that the generated reward structure preserves the optimal policy of the original sparse task.

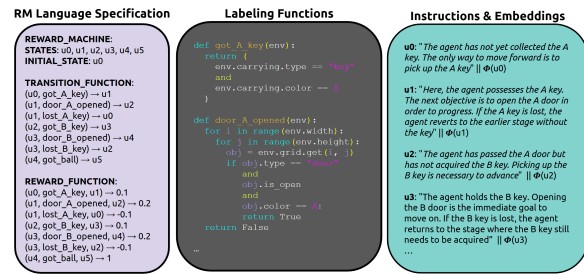

Figure 4: The three core components generated by our method for the `UnlockToUnlock` environment: **(Left)** the RM specification, **(Center)** the labeling functions that drive the state transitions, and **(Right)** the instructions and embeddings for each RM state.

## 3 Empirical Results

We present a series of experiments designed to evaluate the effectiveness and scalability of our method: we test generalization and long-horizon planning in sparse-reward settings with the `MiniGrid` and `BabyAI` suites (Chevalier-Boisvert et al., 2023) (Section 3.1), we evaluate scalability with a resource-gathering task in a 3D, procedurally generated Minecraft world from `Craftium` (Malagón et al., 2024) (Section 3.2), and we demonstrate the applicability of our approach to create RMs that work in continuous control in challenging robotics tasks from `Meta-World` (McLean et al., 2025) (Section 3.3). Finally, we use `XLand-MiniGrid` (Nikulin et al., 2024) to evaluate the generalization capabilities of RL agents trained with LARMs (Section 3.4). Screenshots of these environments are shown in Figure 5 and environment-specific details in Appendix A.2. We used GPT-4o (Hurst et al., 2024) to generate all LARM components for all tasks, with the exception of the 1,000 LARMs for `XLand-MiniGrid`. These were generated using various open-source FMs of different scales for our ablation study.

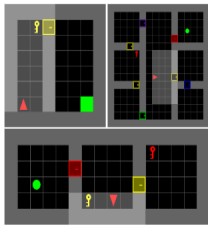 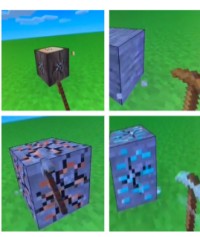 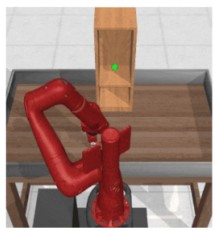 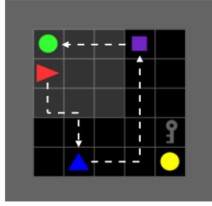

(a) `MiniGrid`      (b) `Craftium`      (c) `Meta-World`      (d) `XLand-MiniGrid`

Figure 5: Evaluation environments. **(a)** `MiniGrid`: tests long-horizon planning with sparse rewards. **(b)** `Craftium`: scaling complexity in a 3D Minecraft-inspired world. **(c)** `Meta-World`: continuous robotic manipulation. **(d)** `XLand-MiniGrid`: tests multi-task generalization.

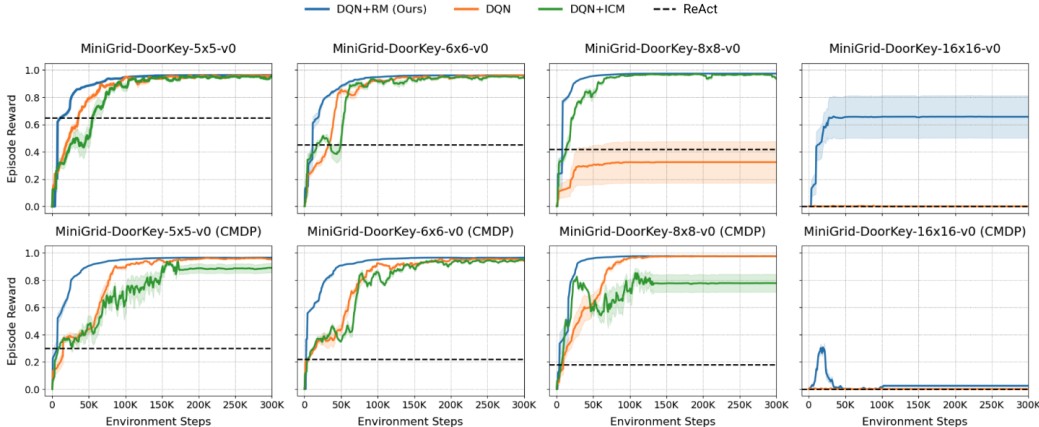

Figure 6: Performance on `MiniGrid-DoorKey` environments of increasing size. The top row shows performance on a fixed map layout, while the bottom row shows performance on procedurally generated layouts. Our method consistently achieves higher rewards across all tasks.

We report results averaged over 3 independent random seeds, with shaded regions and error bars indicating one standard deviation. Comprehensive details for all environments as well as additional results are presented in Appendix A.2, details on the baselines used and additional ablations in A.7, and hyperparameters in A.11.

## 3.1 SPARSE REWARD TASKS

We first evaluate our method on the `MiniGrid` suite of environments, which are challenging due to sparse rewards. For these experiments, we use DQN (Mnih et al., 2013) as the base agent and compare our method (DQN+RM) with a baseline with intrinsic motivation (DQN+ICM) (Pathak et al., 2017), ReAct (Yao et al., 2023) – an LLM-as-agent baseline which generates reasoning traces as it acts in the environment (Paglieri et al., 2024) – as well as an unmodified DQN. In Appendix A.7.4 we include results comparing to a VLM-as-reward-model baseline proposed by Rocamonde et al. (2023). The LARMs used to train our DQN+RM agent are shown in Appendix A.9.

**Our agent successfully solves a suite of challenging exploration tasks where all baselines fail**. We first present results on the `DoorKey` task in Figure 6, showing performance across increasing grid sizes. Our method consistently outperforms the baselines in all settings, including fixed maps (top row) and procedurally generated maps where the layout is randomized each episode (bottom row). We provide an additional analysis of our method's success in Appendix A.7.2. To further test our agent, we select three significantly harder tasks that require longer planning horizons: `UnlockToUnlock`, `BlockedUnlockPickup`, and `KeyCorridor`. As demonstrated in Figure 7, our agent is the only one capable of solving all three tasks and achieving near-perfect reward, while all other baselines fail to make any progress.

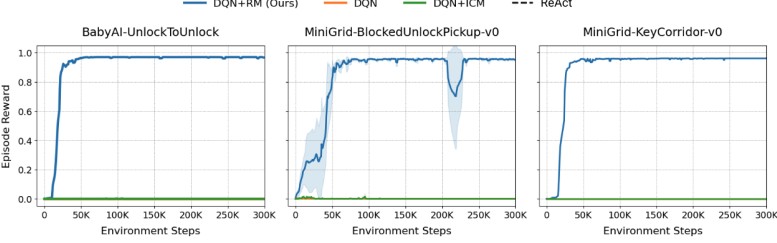

Figure 7: Performance on complex, long-horizon `MiniGrid` tasks. **Our method successfully solves all three, while baselines show no learning**.

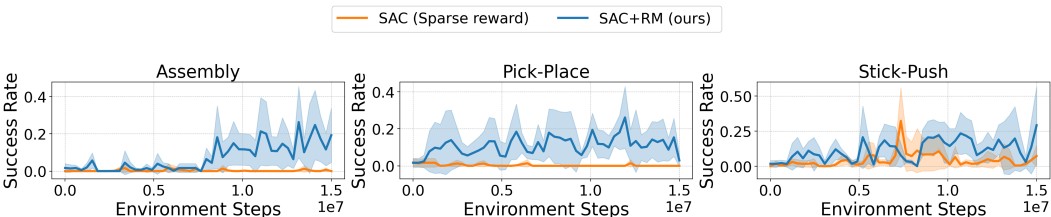

Figure 8: Performance on `Meta-World` manipulation tasks. In most tasks, our method achieves high success rates compared to the sparse reward agent.

## 3.2 SCALING TO COMPLEX 3D ENVIRONMENTS

We assess our method's performance in a complex, procedurally generated 3D environment using `Craftium` (Malagón et al., 2024), which is a Minecraft-based resource-gathering task. Here, the agent's goal is to mine a diamond by first navigating the world to gather the required wood, stone, and iron. The environment provides a sparse reward only upon collecting the final diamond. For this set of experiments we use PPO (Schulman et al., 2017) as the base agent.

As shown in Figure 1 (right), PPO augmented with our generated LARM consistently completes the entire task sequence, while the baseline PPO agent makes minimal progress. This result is particularly significant, as RMs often require manual, expert-driven design which can become challenging in complex, open-ended environments (Icarte et al., 2022). In contrast, we demonstrate the successful application of a RM that is **not only automatically generated by a FM but is also highly effective in a complex 3D, procedurally generated environment**. The LARM achieves this by effectively decomposing the high-level goal into progressive subtasks and providing crucial intermediate rewards. This experiment highlights our framework's ability to handle increased action dimensionality and visual complexity, and it showcases the capability of FMs to leverage their knowledge to automate task decomposition.

## 3.3 ROBOTIC MANIPULATION

**Our framework can automate the complex task of reward engineering for robotic manipulation, providing dense supervision with a FM-generated LARM**. We evaluate this capability in continuous control domains using the `Meta-World` benchmark (McLean et al., 2025), where designing dense reward functions typically requires extensive hand-engineering of low-level signals (e.g., joint angles). Our approach bypasses this difficulty entirely. The resulting reward machine offers richer learning signals than sparse rewards, enabling the agent to make more progress. As demonstrated in Figure 8, our method achieves higher success rates than learning from sparse rewards alone, using SAC (Haarnoja et al., 2018) as the base agent. Additional experiments on `Meta-World` are provided in Appendix A.2.3.

## 3.4 GENERALIZATION THROUGH LANGUAGE EMBEDDINGS

A key design choice in our framework is to condition the agent's policy on the language embeddings of the current RM state, which contrasts with prior work that uses separate policies that do not permit knowledge sharing (Alsadat et al., 2025). In the following, we (i) ablate the roles of the LARM rewards and state embeddings in enabling an RL agent to learn robust, multi-task policies; and (ii) demonstrate how the compositional structure of LARMs leads to zero-shot generalization on unseen tasks. For clarity, we refer here to zero-shot generalization across novel task compositions within the same domain, rather than cross-domain transfer.

**Both structured rewards and language-based state conditioning are essential for learning a robust, multi-task policy.** To disentangle the benefits of the LARM's reward structure from the state embeddings, we conduct an ablation study in a multi-task setting. We train a single Rainbow DQN (Hessel et al., 2018) agent on an increasing number of simultaneous `XLand-MiniGrid` tasks and measure its average success rate. As shown in Figure 9, the baseline agent fails to generalize as the number of tasks increases. Providing the policy with only the state embeddings gives it a weak learning signal that degrades quickly. Conversely, providing only the LARM rewards enables

multi-task learning, but the policy struggles as it is unaware of the active sub-goal. Our full method, which uses both the dense rewards from the LARM and the state embeddings to condition the policy, is robust and maintains high performance even when trained on 10 simultaneous tasks.

**The compositional structure of LARMs enables zero-shot generalization to novel tasks composed of previously seen sub-goals.** The ultimate test of our compositional approach is whether the trained policy, $\pi(a_t|s_t, z_{u_t})$, can solve a novel task without any additional training. We design an experiment where $\pi$ is trained on a set of tasks, $\{\mathcal{T}_A, \mathcal{T}_B\}$, each with an associated LARM, $\mathcal{R}_A$ and $\mathcal{R}_B$. During training, the policy learns skills corresponding to the union of all sub-goal embeddings, $\{z_u|u \in U_A \cup U_B\}$. At evaluation time, we introduce a new, unseen task, $\mathcal{T}_C$, with a novel LARM, $\mathcal{R}_C$, generated by the FM. Zero-shot success is possible if the set of sub-goals in the new task is composed of elements semantically familiar from training, i.e., if for any state $u' \in U_C$, its embedding $z_{u'}$ is close to an embedding seen during training.

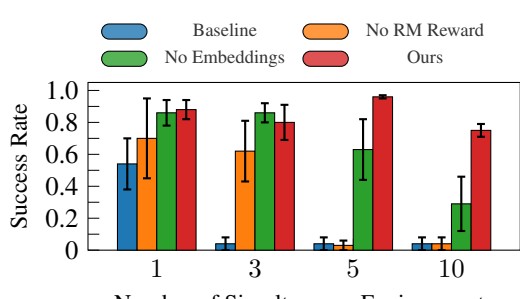

Figure 9: Ablation study of the components of ARM-FM. A Rainbow agent is trained on an increasing number of tasks. While baselines fail to generalize, **only our full method (combining LARM rewards and state embeddings) maintains high success as the number of tasks grows**.

As illustrated in Figure 10, the agent successfully solves Task C. When the LARM for Task C transitions to a state $u'_t$, the policy receives the input $(s_t, z_{u'_t})$. Because the embedding $z_{u'_t}$ (e.g., for "Pick up a blue key, Position yourself to the right of the blue pyramid") is already located in a familiar region of the skill space, the policy can reuse the relevant learned behavior to make progress and solve the unseen composite task.

## 4 ARM-FM: IN-DEPTH ANALYSIS

We now conduct a fine-grained analysis of our method's key components by evaluating (i) the quality of the LARMs generated by different FMs; and (ii) the semantic structure of the state embeddings.

**Larger foundation models generate syntactically correct task structures with significantly higher reliability.** To evaluate the FM's generation capabilities, we sampled 1,000 diverse tasks from the `XLand-MiniGrid` environment (Nikulin et al., 2024) and prompted various open-source models to generate the corresponding reward machines, Python labeling functions, and natural language instructions. We compare models of different scales from the `Qwen` (Qwen, 2025), `Gemma` (Gemma, 2025), `Llama` (Dubey et al., 2024) and `Mistral` families. We employed an LLM-as-judge protocol to score the correctness of the generated artifacts (Gu et al., 2024), using *Qwen3-30B-A3B-Instruct-*

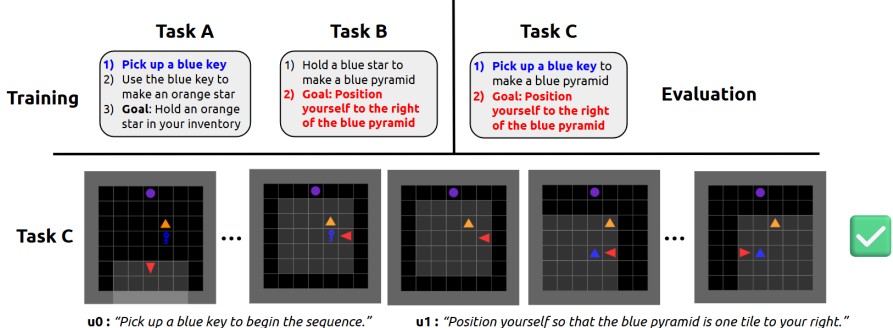

Figure 10: Demonstration of zero-shot generalization. An agent is trained on a set of tasks (A, B). At evaluation, it is given a new LARM for an unseen composite task (C). Because the sub-tasks in C (e.g., "Pick up a blue key") are semantically familiar from training, the agent can reuse learned skills to solve the novel task without any fine-tuning.

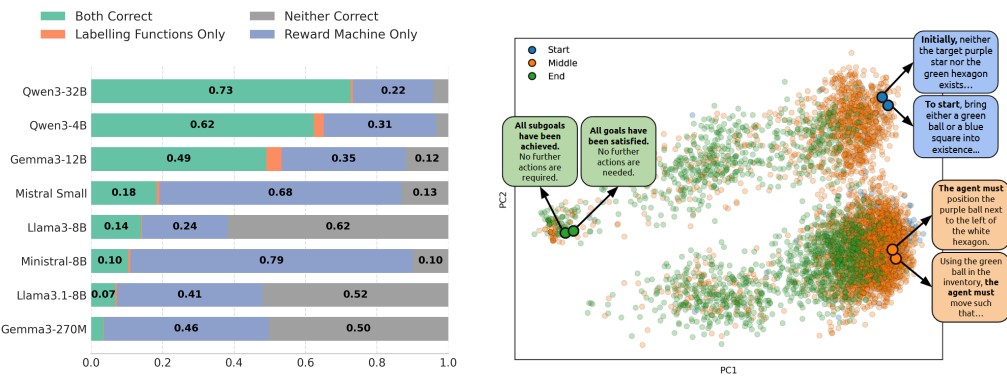

(a) FM generation correctness.

(b) Semantic structure of state embeddings.

Figure 11: Analysis of FM-generated task components. **(a)** An LLM-as-judge evaluation across 1,000 tasks reveals a strong scaling trend, where larger foundation models more reliably generate correct RM structures and verifier code. **(b)** PCA visualization of thousands of state instruction embeddings shows a clear semantic structure, with start, middle, and end states forming distinct clusters.

*2507* as the judge (a FM not used to generate the LARMs). Figure 11a shows a clear scaling trend: larger models like *Qwen3-32B* are significantly more capable of generating fully correct task specifications. Interestingly, some models exhibit different strengths; for instance, *Mistral-Small* is more adept at generating a valid RM structure than correct labeling code, highlighting the distinct reasoning and coding capabilities required.

**The FM-generated state instructions produce a semantically coherent embedding space that clusters related sub-goals.** Beyond syntactic correctness, the agent's ability to generalize depends on the semantic quality of the state instruction embeddings. A well-structured embedding space should group semantically similar sub-tasks together, regardless of the overarching task. We analyze this by visualizing the embeddings of state instructions from the 1,000 generated `XLand-MiniGrid` tasks using PCA (Abdi & Williams, 2010), as shown in Figure 11b. We used *Qwen3-30B-A3B-Instruct-2507* to obtain the embeddings. The embeddings form distinct and meaningful clusters, with instructions corresponding to the start, middle, and end of a task occupying different regions of the space. Notably, instructions with similar meanings from different tasks cluster together, confirming that the FM produces a coherent representation. This underlying semantic structure is what enables a shared policy to treat related sub-tasks in a similar manner, forming the foundation for skill transfer.

## 5 RELATED WORK

**Reward Machines in Reinforcement Learning.** RMs are a formal language representation of reward functions that expose the temporal and logical structure of tasks, thus enabling decomposition, transfer, and improved sample efficiency in learning (Icarte et al., 2018; 2022). For these reasons, RMs and related formal methods for task specification have been applied to address diverse challenges, from multiagent task decomposition (Neary et al., 2021; Smith et al., 2023) to robotic manipulation and task planning (Camacho et al., 2021; He et al., 2015; Cai et al., 2021). Recent work continues to broaden their applicability, by studying extensions that increase their expressivity (Varricchione et al., 2025), and by addressing uncertainty in symbol grounding and labeling functions (Li et al., 2024; 2025). While RMs can be difficult to design for non-experts, Toro Icarte et al. (2019) and Xu et al. (2020) propose methods that simultaneously learn RMs and RL policies, if the RM is unknown a priori.

Recent work also explores FM-driven automata. While some approaches treat RM states as isolated symbols Alsadat et al. (2025), requiring careful state mapping for policy re-use, others use FMs with classic algorithms for automaton discovery Vazquez-Chanlatte et al. (2025), which requires expert demonstrations. Our work differs by generating RMs directly from language descriptions, without behavioral examples. Concurrently, methods like RAD embeddings Yalcinkaya et al. (2024) have been proposed to condition the policy on the automaton's topology. We take a complementary,

language-first approach: our FM generates language-aligned embeddings for the meaning of each state, which our results show effectively grounds the policy.

By contrast, ARM-FM not only generates RMs from natural language task descriptions, but also introduces a natural mechanism for connecting RM states by embedding their associated subtask descriptions in a shared latent space. Conditioning the policy on these language embeddings can thus enable knowledge transfer across similar subtasks, even when they occur in different RMs.

**Foundation Models in Decision-Making.** The emergence of FMs has inspired two main lines of work in sequential decision-making. The first uses FMs directly as autonomous agents (Paglieri et al., 2024). Approaches such as ReAct, (Yao et al., 2023) Voyager (Wang et al., 2023) and SayCan Ahn et al. (2022) employ large language models (LLMs) to perform reasoning, planning, and acting. While these systems demonstrate strong capabilities in complex domains, they heavily depend on environment abstractions (e.g., textual interfaces or code as actions) that bypass many of the low-level perception and control challenges central to RL. In contrast, we use RMs to structure policies for learning agents, solving complex sparse reward tasks beyond the reach of non-learning, in-context methods like ReAct which additionally require high-level textual interfaces.

A second line of research integrates FMs with RL training by using them to provide auxiliary signals such as high-level goals or reward feedback. For example, Motif (Klissarov et al., 2023) elicits trajectory-level preferences from FMs and distills them into a reward model. ONI (Zheng et al., 2024) aggregates asynchronous LLM feedback into a continuously updated reward function. Eureka (Ma et al., 2023) leverages evolutionary strategies to generate programmatic reward functions, which are then used to train downstream policies. ELLM uses pretrained LLMs to suggest plausibly useful goals and trains RL agents with goal-reaching rewards. These approaches illustrate the potential of injecting FM knowledge to shape RL objectives. However, the outputs are typically limited to an opaque reward model, rather than a structured, compositional representation of the task.

Our work differs in the structure of the FM–RL interface. We employ FMs to generate language-aligned RMs: structured, compositional, and interpretable representations of task reward functions. This formulation combines the expressivity of FMs with the explicit, modular decomposition, and human-in-the-loop refinement enabled by RMs, offering a principled path toward hierarchical and interpretable RL. Additionally, our method does not depend on specific environment abstractions (Wang et al., 2023) or the availability of a natural language interface (Klissarov et al., 2023). We provide a detailed comparison with existing methods in Section A.6 (see Table 4).

## 6 CONCLUSION

In this work, we introduce Automated Reward Machines via Foundation Models (ARM-FM), a framework that bridges the critical gap between the semantic reasoning of foundation models and the low-level control of reinforcement learning agents. Our central contribution is a method for automatically generating Language-Aligned Reward Machines (LARMs) from natural language. We demonstrated that by conditioning a single policy on the embeddings of the LARM's natural language state descriptions, we transform the reward machine from a static plan into a compositional library of reusable skills. Our experiments confirmed the effectiveness of this approach. We showed that ARM-FM solves a suite of long-horizon, sparse-reward tasks across diverse domains – from 2D grid worlds to a procedurally generated 3D crafting environment – that are intractable for strong RL baselines. Our analysis revealed that this performance is underpinned by a coherent semantic structure in the state embedding space and that both the structured rewards and the state embeddings are critical for robust multi-task learning. The ultimate validation of our compositional approach was the demonstration of zero-shot generalization to a novel, unseen task without any additional training.

Ultimately, this work establishes language-aligned reward machines as a powerful and versatile framework connecting foundation models, RL agents, and human operators. The modular, language-based structure allows FMs to generate accurate plans, agents to learn generalizable skills, and humans to easily inspect and refine the task specifications. While this paradigm is promising, one tradeoff of our approach is the human verification step during RM generation. On one hand, this step may be viewed as a feature – the language-based reward structures output by ARM-FM provide an interface for humans to interpret and refine task specifications. On the other hand, this step presupposes access to human verifiers. We note, however, that such verifiers are not strictly required,

although they can improve output quality when available. Furthermore, future work may reduce or eliminate this dependence by exploiting the automaton-based structure of RMs to enable automated self-correction, for example, through formal verification. More broadly, we believe this work paves the way for a new class of RL agents that can translate high-level human intent and FM-generated plans into competent, generalizable, and interpretable behavior.

## ACKNOWLEDGEMENTS

We want to acknowledge funding support from Natural Sciences and Engineering Research Council (NSERC) of Canada, Google Research, and The Canadian Institute for Advanced Research (CIFAR) and compute support from Digital Research Alliance of Canada, Mila IDT, and NVidia.

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

# A  APPENDIX

## A.1  LLM USAGE

In the preparation of this manuscript, we used large language models (LFMs) as writing assistants. Their role was strictly limited to improving the grammatical correctness of our text.

The LLM was prompted to review author-written drafts and provide feedback on phrasing or flag passages that were potentially unclear. No standalone text was generated by the LLM for inclusion in the paper. All core scientific ideas, experimental results, and analyses are the original work of the human authors, who take full responsibility for the final content.

## A.2  ENVIRONMENT DETAILS

### A.2.1  MINIGRID AND BABYAI ENVIRONMENTS

This section provides a detailed description of the MiniGrid and BabyAI environments used in our experiments. These tasks are selected to test distinct agent capabilities, ranging from basic exploration and generalization to complex, long-horizon planning and reasoning.

In the **DoorKey** task, the agent must find a key within the observable room, use it to unlock a door, and navigate to a goal location. The sparse reward, given only upon reaching the goal, makes this a classic exploration challenge. We use procedurally generated versions of this task to evaluate generalization to novel map layouts.

The **BlockedUnlockPickup** task significantly increases the planning complexity. The agent must first move a blocking object (a ball), retrieve a key from the main room, unlock a door, and finally pick up a target box in a separate room. This requires a long and precise sequence of actions to solve.

**UnlockToUnlock** is a BabyAI task that tests hierarchical reasoning and memory. The agent must find a key for a first door to navigate to a different room, which in turn contains a key for a second, final door to the goal room. This creates a nested dependency structure with extremely sparse rewards, making it exceptionally difficult.

The **KeyCorridor** environment is a difficult exploration task. The agent starts in a corridor with multiple rooms, one of which contains a hidden key. It must explore the side rooms to find the key, return to the corridor to unlock the correct door, and reach the final goal.

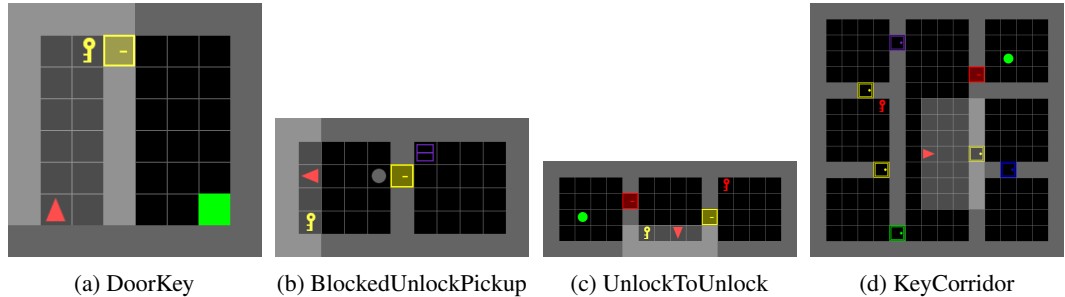

(a) DoorKey    (b) BlockedUnlockPickup    (c) UnlockToUnlock    (d) KeyCorridor

### A.2.2  CRAFTIUM

Craftium is a high-performance, open-source 3D voxel platform designed for reinforcement learning research. Inspired by Minecraft, Craftium offers rich, procedurally generated open worlds and fully destructible environments. Built on the C++-based Luanti engine, it provides significant performance advantages over Java-based alternatives and integrates natively with modern RL frameworks through the Gymnasium API. This makes it an ideal testbed for assessing agent performance on tasks requiring generalization in visually complex, high-dimensional settings.

Within this platform, we designed a challenging open-world task where the agent's sole objective is to mine a diamond. The environment is procedurally generated for each episode, and a sparse reward

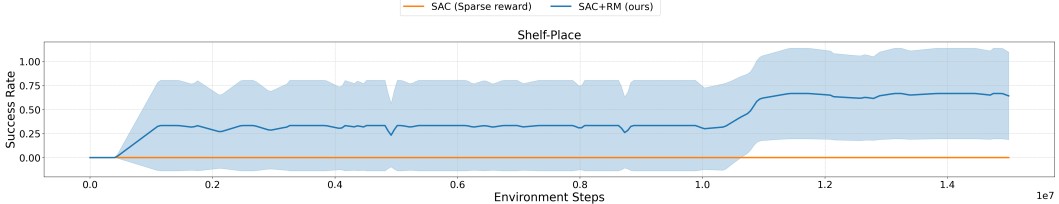

Figure 13: Performance on the Meta-World `Shelf-Place` task: With careful hyperparameter tuning, an agent that maximizes the sum of sparse task and reward machine rewards significantly outperforms the sparse reward agent.

is only awarded upon successful collection of the diamond. This task implicitly requires a long and complex sequence of actions: gathering wood, stone, iron and diamond in this order.

As shown in Figure 1 (Right), a baseline PPO agent fails to learn a meaningful policy and makes negligible progress on the task. In contrast, PPO augmented with our generated reward machine consistently learns the full sequence of behaviors required to solve the task. This result demonstrates that our framework effectively scales to visually complex, procedurally generated 3D environments with extremely sparse rewards.

### A.2.3 META-WORLD

We evaluate our method on a subset of Meta-World, a robotic manipulation benchmark originally created for evaluating multi-task and meta-RL algorithms. We adapted this benchmark to our setting by replacing the dense reward with a sparse reward signal, and we compare an agent that maximizes only the sparse reward signal to an agent that maximizes the sum of the sparse reward and the reward from the reward machine (Figure 14). We evaluated our method on the following tasks:

- `Assembly`: The agent task is to pick a nut and place into a peg
- `Bin-Picking`: The agent's task is to pick a puck from one bin and place it in another bin.
- `Pick-Place`: The task is to pick a puck and place it in a specific goal location.
- `Shelf-Place`: The agent's task is to pick a puck and place it on a shelf.
- `Stick-Push`: The agent's task is to grab a stick and push a box using the stick.

The observation and action spaces share the same structure among the tasks. The observation vector consists of the robot's end-effector 3D coordinates, a scalar value indicating whether the gripper is open or closed, and the position and orientation information of objects in the environment. At each time step, the current observation is concatenated with the observation from the previous time step, along with the goal position, resulting in a 39-dimensional vector. The action vector consists of three displacement values ($dx, dy,$ and $dz$) of the end effector, with an additional action for opening or closing the gripper.

The result of the main experiment is shown in Figure 8. We also show in Figure 13 that, with more careful hyperparameter tuning, the agent augmented with the reward from the reward machine can solve the task with a high success rate. Moreover, the reward machine can be combined with off-the-shelf intrinsic exploration rewards, such as RND (Figure 15). This results in overall better performance in most environments compared to the results in Figure 8.

### A.2.4 XLAND-MINIGRID

To evaluate our agent's generalization capabilities and its ability to adapt to novel situations, we use the `XLand-MiniGrid` benchmark. This suite of environments is specifically designed for meta-reinforcement learning research, combining the procedural diversity and depth of DeepMind's XLand with the minimalism and fast iteration of MiniGrid.

The entire framework is implemented from the ground up in JAX, a design choice that enables massive parallelization and makes large-scale experimentation accessible on limited hardware. Its

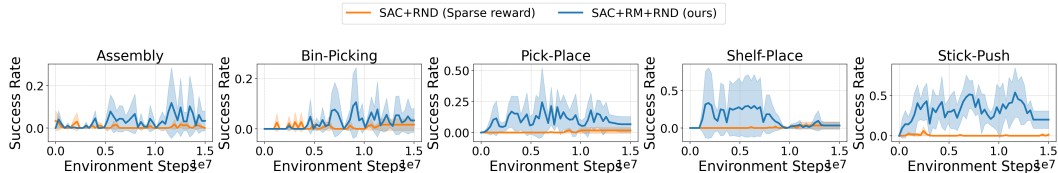

Figure 14: Performance on the Meta-World on five tasks, our method offers richer reward signal than sparse reward.

Figure 15: Performance on the Meta-World: When combining the reward machine with the RND exploration term, our method can make use of exploration bonuses, resulting in better overall performance.

core feature is a compositional system of rules (e.g., "keys open doors of the same color") and goals (e.g., "go to the blue box") that can be arbitrarily combined to procedurally generate a vast and diverse distribution of distinct tasks. This allows for the creation of structured curricula and rigorous tests of an agent's ability to infer the underlying rules of a new environment and adapt its strategy accordingly.

In our experiments in Section 4, we leverage `XLand-MiniGrid` to assess how effectively our framework can adapt across this wide distribution of tasks. The primary challenge in this setting is not to master a single, static task, but to develop a policy that can quickly recognize the objectives and constraints of a newly sampled environment and formulate a successful plan on the fly. This makes it a powerful testbed for evaluating the adaptability and generalization of our approach.

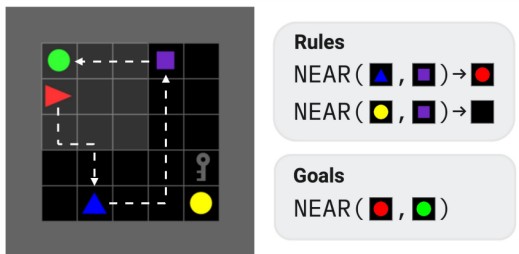

Figure 16: A sample task from our XLand-MiniGrid distribution, with the optimal solution path highlighted. The agent must infer that placing the blue pyramid near the purple square creates a red circle, which must then be moved to the green goal. A distractor object (yellow circle) can render the task unsolvable. The agent is unaware of these rules, and object positions are randomized to test for adaptation.

XLand-MiniGrid provides a formal language for procedurally generating tasks from a combination of goals, rules, and initial object placements. This allows for the creation of a vast and diverse task space. The complete sets of supported goals and rules, adapted from the original XLand-MiniGrid paper, are detailed in Tables 1 and 2. Figure 17 illustrates our framework's zero-shot generalization capabilities within this formal language, mirroring the experiment from Figure 10. An agent trained on tasks A and B can successfully solve the novel composite Task C, demonstrating its ability to understand and execute policies based on the underlying formal structure of the environment.

Table 1: Supported goals in the XLand-MiniGrid formal language.

| Goal | Meaning | ID |
|---|---|---|
| EmptyGoal | Placeholder goal, always returns False | 0 |
| AgentHoldGoal(a) | Whether agent holds a | 1 |
| AgentOnTileGoal(a) | Whether agent is on tile a | 2 |
| AgentNearGoal(a) | Whether agent and a are on neighboring tiles | 3 |
| TileNearGoal(a, b) | Whether a and b are on neighboring tiles | 4 |
| AgentOnPositionGoal(x, y) | Whether agent is on (x, y) position | 5 |
| TileOnPositionGoal(a, x, y) | Whether a is on (x, y) position | 6 |
| TileNearUpGoal(a, b) | Whether b is one tile above a | 7 |
| TileNearRightGoal(a, b) | Whether b is one tile to the right of a | 8 |
| TileNearDownGoal(a, b) | Whether b is one tile below a | 9 |
| TileNearLeftGoal(a, b) | Whether b is one tile to the left of a | 10 |
| AgentNearUpGoal(a) | Whether a is one tile above agent | 11 |
| AgentNearRightGoal(a) | Whether a is one tile to the right of agent | 12 |
| AgentNearDownGoal(a) | Whether a is one tile below agent | 13 |
| AgentNearLeftGoal(a) | Whether a is one tile to the left of agent | 14 |

Table 2: Supported rules in the XLand-MiniGrid formal language.

| Rule | Meaning | ID |
|---|---|---|
| EmptyRule | Placeholder rule, does not change anything | 0 |
| AgentHoldRule(a) $\rightarrow$ c | If agent holds a replaces it with c | 1 |
| AgentNearRule(a) $\rightarrow$ c | If agent is on neighboring tile with a replaces it with c | 2 |
| TileNearRule(a, b) $\rightarrow$ c | If a and b are neighboring tiles, replaces one with c and removes the other | 3 |
| TileNearUpRule(a, b) $\rightarrow$ c | If b is one tile above a, replaces one with c and removes the other | 4 |
| TileNearRightRule(a, b) $\rightarrow$ c | If b is one tile to the right of a, replaces one with c and removes the other | 5 |
| TileNearDownRule(a, b) $\rightarrow$ c | If b is one tile below a, replaces one with c and removes the other | 6 |
| TileNearLeftRule(a, b) $\rightarrow$ c | If b is one tile to the left of a, replaces one with c and removes the other | 7 |
| AgentNearUpRule(a) $\rightarrow$ c | If a is one tile above agent, replaces it with c | 8 |
| AgentNearRightRule(a) $\rightarrow$ c | If a is one tile to the right of agent, replaces it with c | 9 |
| AgentNearDownRule(a) $\rightarrow$ c | If a is one tile below agent, replaces it with c | 10 |
| AgentNearLeftRule(a) $\rightarrow$ c | If a is one tile to the left of agent, replaces it with c | 11 |

For the experiments in Section 4, we evaluate performance on the first 1,000 tasks from the *medium-1m* benchmark in XLand-MiniGrid. The specific seeds visualized in Figure 9 are: 197 (1-task); 212, 197, 260 (3-task); 212, 197, 260, 859, 594 (5-task); and 212, 197, 260, 859, 594, 571, 602, 751, 660, 616, for the 10-task setting.

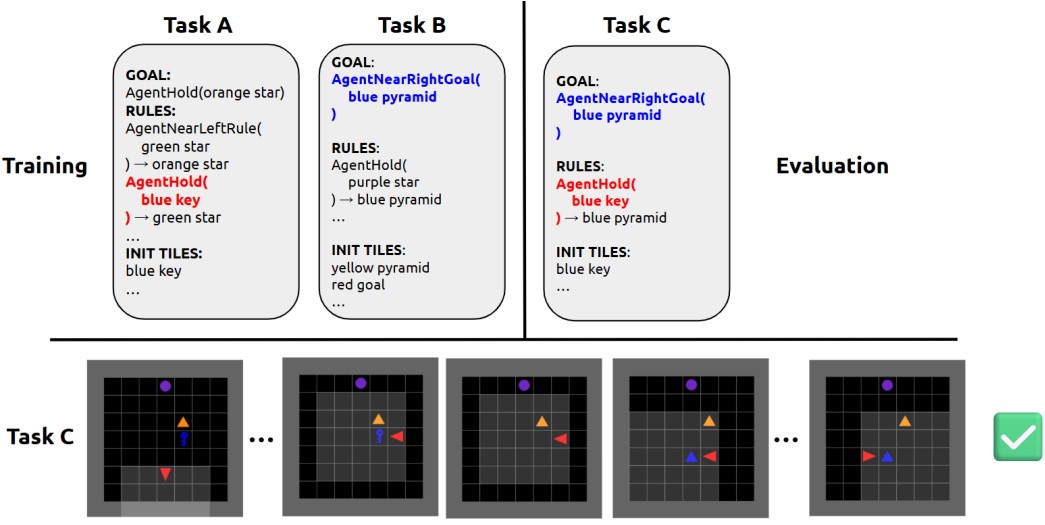

Figure 17: Demonstration of zero-shot generalization. An agent is trained on a set of tasks (A, B). At evaluation, it is given a new reward machine for an unseen composite task (C). Because the sub-tasks in C (e.g., "Pick up a blue key") are semantically familiar from training, the agent can reuse learned skills to solve the novel task without any fine-tuning.

### A.3 DQN TRAINING WITH LARMS

This section provides a detailed description of the reinforcement learning training procedure used in our work. We adapt the standard Deep Q-Network (DQN) algorithm (Mnih et al., 2013) to incorporate Language-Aligned Reward Machines (LARMs). The core idea is to augment the agent's state representation with the current state of the LARM and to use the LARM to provide a dense, structured reward signal.

Algorithm 1 formalizes this process. The key modifications to the standard DQN algorithm are highlighted in blue. These changes include:

1. **Augmented State Input**: The policy, represented by the Q-network, takes as input not only the environment state $s_t$ but also the language embedding of the current LARM state, $\phi(u_t)$. This allows the agent to learn state- and task-dependent skills.

2. **LARM State Transitions**: After each environment step, the LARM is updated based on the new environment state $s_{t+1}$ and the action $a_t$ taken. The labeling function $\mathcal{L}$ determines if a relevant event occurred, which in turn may cause a transition to a new LARM state $u_{t+1}$.

3. **Combined Reward Signal**: The agent learns from a composite reward signal that is the sum of the base environment reward $R_t$ and the reward from the LARM, $R_t^{\text{RM}}$. This provides dense, incremental feedback for completing subtasks.

4. **Augmented Experience Replay**: The transitions stored in the replay memory $\mathcal{D}$ are augmented to include the LARM states, i.e., $(s_t, u_t, a_t, R_t^{\text{total}}, s_{t+1}, u_{t+1})$. This ensures the agent learns the Q-values over the joint state space.

By conditioning the policy on semantic embeddings of LARM states, the agent can effectively generalize across related subtasks, leading to improved sample efficiency and performance on complex, long-horizon tasks.

---

**Algorithm 1** DQN Training with Language-Aligned Reward Machines (LARMs)

---

1: **Initialize:** Replay memory $\mathcal{D}$ to capacity $N$.
2: **Initialize:** Q-network $Q$ with random weights $\theta$.
3: **Initialize:** Target Q-network $\hat{Q}$ with weights $\theta^- \leftarrow \theta$.
4: **Initialize:** Update frequency $C$ for the target network.
5: **Input:** LARM $(U, u_I, \delta, R, \mathcal{L})$ from Section 2.1.
6: **Input:** State instruction embedding function $\phi(\cdot)$.
7: **for** episode = 1 to M **do**
8:     Reset environment to get initial state $s_0$.
9:     Reset LARM to its initial state, $u_0 \leftarrow u_I$.
10:     **for** t = 0 to T-1 **do**
11:         With probability $\epsilon$, select a random action $a_t$.
12:         Otherwise, select $a_t = \arg\max_a Q(s_t, \phi(u_t), a; \theta)$.
13:         Execute action $a_t$ in the environment, observe reward $R_t$ and next state $s_{t+1}$.
14:         Get LARM event via labeling function: $e_t = \mathcal{L}(s_{t+1}, a_t)$.
15:         Get next LARM state: $u_{t+1} = \delta(u_t, e_t)$.
16:         Get LARM reward: $R_t^{\text{RM}} = R(u_t, e_t)$.
17:         Compute total reward: $R_t^{\text{total}} = R_t + R_t^{\text{RM}}$.
18:         Store transition $(s_t, u_t, a_t, R_t^{\text{total}}, s_{t+1}, u_{t+1})$ in $\mathcal{D}$.
19:         Sample a random minibatch of transitions $(s_j, u_j, a_j, R_j^{\text{total}}, s_{j+1}, u_{j+1})$ from $\mathcal{D}$.
20:         Set target $y_j = \begin{cases} R_j^{\text{total}} & \text{if episode terminates at step } j+1 \\ R_j^{\text{total}} + \gamma \max_{a'} \hat{Q}(s_{j+1}, \phi(u_{j+1}), a'; \theta^-) & \text{otherwise} \end{cases}$
21:         Perform a gradient descent step on $(y_j - Q(s_j, \phi(u_j), a_j; \theta))^2$.
22:         Every $C$ steps, update the target network: $\theta^- \leftarrow \theta$.
23:     **end for**
24: **end for**

---

### A.4    HUMAN-IN-THE-LOOP LARM GENERATION

In Figure 3, we show the self-improvement loop used to generate reward machines, where we instantiate both generator and critic foundation models to iteratively refine the LARMs. A key advantage of our LARM framework is that the interface to define and refine them is natural language, which allows human operators to easily interpret and intervene in the generation process. This section provides a transparent breakdown of the specific human-in-the-loop efforts involved for each environment presented in this paper, summarized in Table 3. To facilitate this, we implemented an interactive interface where a human operator could replace the critic foundation model during any round of self-improvement. In this mode, the generator model would receive the full history of LARM attempts and critic feedbacks, followed by a new refinement comment provided directly by the human. This design allowed us to seamlessly integrate both FM-generated and human-provided feedback within the same improvement loop.

Table 3: Summary of Human-in-the-Loop Effort for LARM Generation.

| Environment | Human? | Description of Intervention |
|---|---|---|
| MiniGrid-DoorKey (all sizes) | ✗ | No intervention. The FM self-improvement loop was sufficient. Human check confirmed correctness after 3 iterations. |
| MiniGrid-UnlockPickup | ✓ | **Yes.** The initial LARM missed an edge case: the agent dropping a key after pickup. A human provided feedback to add this transition (reflecting a loss of progress). The FM incorporated this, and the task was solved. |
| MiniGrid-BlockedUnlockPickup | ✗ | No intervention. The FM self-improvement loop was sufficient. |
| MiniGrid-KeyCorridorS3R3 | ✓ | **Yes.** The originally generated LARM was too sparse. A human provided high-level advice to "define intermediate rewards" and suggested "crossing doors" or "entering new rooms" as progress signals. The FM then generated a denser, effective LARM. |
| Craftium | ✗ | No intervention. This was notable, as the FM successfully leveraged its latent knowledge of Minecraft-like game mechanics without guidance. |
| XLand-MiniGrid | ✗ | No human intervention on any of the 1,000 generated LARMs. Correctness was validated automatically using the LLM-as-judge method (as shown in Figure 11, left). |
| MetaWorld (all tasks) | ✓ | **Yes.** The initial reward values in the LARM were leading the agent to a local minima, for example, grasping the object without moving it to the specified location. A human scaled the reward values for specific events in the LARM to avoid the local minima, without changing the events themselves which we assessed appropriate. |

## A.5 THEORETICAL PROPERTIES OF LARM-GUIDED RL

In this section, we formalize the relationship between the original sparse-reward environment and the dense-reward objective created by the LARM. We show that under the conditions met by our generated LARMs, optimizing the LARM-augmented reward preserves the optimal policy of the original task.

**Preliminaries.** As defined in Section 2, let the environment be an MDP $\mathcal{M} = \langle S, A, P, \gamma \rangle$. The original task is defined by a sparse reward function $R_{\text{task}}(s) = R_{\text{goal}}$ if $s \in S_{\text{goal}}$ (a terminal state), and 0 otherwise. The LARM is $\mathcal{A} = \langle U, u_0, F, \delta, R_{\text{LARM}} \rangle$, where $\delta : U \times \mathcal{L} \to U$ is the transition function and $R_{\text{LARM}}$ is the LARM reward function. This induces the cross-product MDP $\mathcal{M}_{\text{LARM}}$, where the agent's reward $r_t$ is determined by $R_{\text{LARM}}$ based on transitions in $\mathcal{A}$.

[Optimality Preservation] *Assume the generated LARM $\mathcal{A}$ contains no positive reward cycles (i.e., for any cycle $u_i \to ... \to u_i$, the sum of rewards $R_{\text{LARM}}$ along the cycle is $\leq 0$). Assume also that the final reward $R_{goal}$ (obtained on transition to an accepting state $u \in F$) is strictly greater than the cumulative reward of any non-terminal trajectory. Then, a policy $\pi^*$ that is optimal for the cross-product MDP $\mathcal{M}_{LARM}$ is also optimal for the original sparse MDP $\mathcal{M}$.*

*Proof Sketch.* The condition of no positive reward cycles is key. It ensures that the value of any non-terminal looping trajectory is bounded and not preferable to progressing toward the goal. Any cycles in the LARM (e.g., for losing progress) must have a non-positive cumulative reward, which prevents the agent from creating reward traps. Because the terminal reward $R_{\text{goal}}$ is set to be strictly dominant, the optimal policy for $\mathcal{M}_{\text{LARM}}$ will always maximize value by finding a path to an accepting state $u \in F$. The intermediate rewards from $R_{\text{LARM}}$ thus act as potential-based shaping to guide exploration, densifying the sparse signal without altering the set of optimal policies.

This proposition holds for the LARMs used in this paper (see Appendix A.9). For instance, the LARM for `UnlockPickup` contains cycles, such as losing a key (`(u1, lost_y_key) -> u0`). However, this transition has a negative reward (`-0.1`) that exactly cancels the positive reward from acquiring the key (`+0.1`). This "potential-based" structure ensures no positive cycles are created, satisfying the proposition's condition. The `DoorKey` LARM contains a similar zero-sum cycle. The `Craftium` LARM is a Directed Acyclic Graph and thus trivially satisfies the condition. All other LARMs used in our experiments adhere to this property.

## A.6 COMPARISON WITH RELATED WORK

To further clarify our contributions, we provide a detailed comparison with prior work in Table 4. The table is split into two categories: (1) methods that use FMs to synthesize or interact with automata and (2) general FM-guided RL frameworks. This comparison highlights that ARM-FM is unique in its ability to directly generate a complete, semantically-grounded automaton from language without requiring expert demonstrations, and then use it to train a learning agent.

Table 4: Comparison of ARM-FM with FM-driven automata and FM-guided RL frameworks. Our method's advantages are highlighted in **bold**.

| Method | Generates RM? | Requires Demos? | Agent Type | Key Assumption | Primary FM Output / Role |
|---|---|---|---|---|---|
| *FMs for Automata Synthesis* | | | | | |
| L*LM (Vazquez-Chanlatte et al., 2025) | Yes | **Yes** | No agents trained | Expert Demonstrations | Answers membership queries for the L* algorithm. |
| RAD (Yalcinkaya et al., 2024) | **No** | No | RL (Learned) | RMs are given | - |
| Alsadat et al. (2025) | Yes | No | RL (Learned) | SAT-based RM learning | FMs generate text as feedback to a SAT-based algorithm to learn RMs |
| **ARM-FM (Ours)** | **Yes** | **No** | **RL (Learned)** | **Language Specification** | **FMs generate LARM + *semantic* embeddings from language end-to-end.** |
| *FM-Guided RL Frameworks* | | | | | |
| ReAct (Yao et al., 2023) | No | No | In-Context (CoT) | Text Interface | Generates text-based Chain-of-Thought reasoning and actions. |
| SayCan (Ahn et al., 2022) | No | **Yes (Skills)** | Pre-trained Skills | Pre-defined Skills | Scores affordances for a set of pre-defined skills. |
| Voyager (Wang et al., 2023) | No | No | In-Context (Code) | Code Interface | Generates Python code for exploration (Minecraft-specific). |
| Eureka (Ma et al., 2023) | No | No | RL (Learned) | Reward Src Code | Evolves the codebase of a programmatic reward function. |
| Motif (Klissarov et al., 2023) | No | No | RL (Learned) | Text Captions | Distills FM-generated trajectory preferences into a reward model. |
| MaestroMotif (Klissarov et al., 2024) | No | No | RL (Learned) | Manually-defined skills | Uses LLM feedback to design rewards for pre-defined skills |
| ELLM (Du et al., 2023) | No | No | RL (Learned) | FM query at each state | Suggests plausibly useful goals based on the agent's current state. |

### A.7 Additional Results

#### A.7.1 MiniGrid - Exploration Baselines

For clarity, the main paper presents results against the best-performing exploration baseline from our evaluation, the Intrinsic Curiosity Module (ICM). In this section, we provide a detailed comparison of the three intrinsic motivation methods we tested: ICM, Random Network Distillation (RND), and Disagreement. All baseline implementations are adapted from the well-tested RLeXplore library (Yuan et al., 2024).

Figure 18 shows the comparative performance of these methods on the DoorKey tasks. The results demonstrate that ICM consistently outperformed the other methods in our tested environments, justifying its selection as the primary exploration baseline for our main analysis.

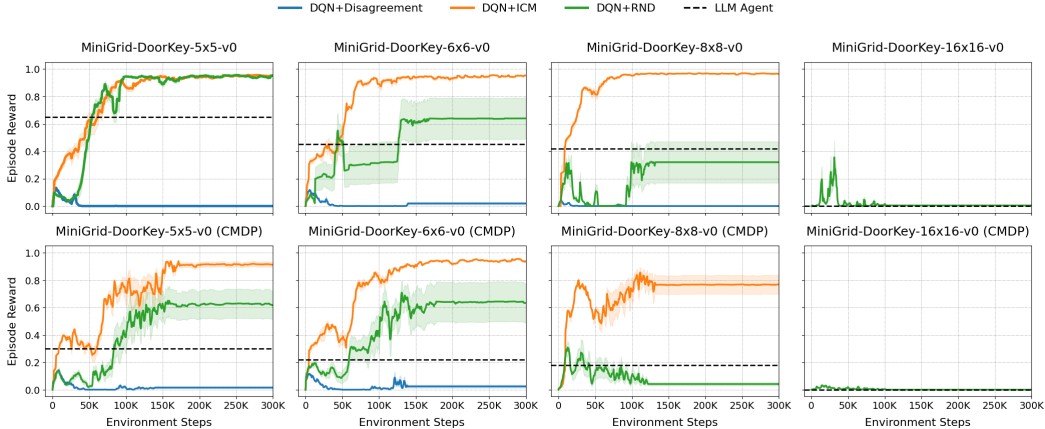

Figure 18: Comparison of exploration baselines (ICM, RND, Disagreement) on the MiniGrid DoorKey tasks. ICM demonstrates the strongest and most consistent performance, establishing it as the most competitive exploration baseline for our experiments.

#### A.7.2 MiniGrid - Analysis of LARM Rewards

This section provides a fine-grained analysis of how LARM-generated rewards guide an agent toward solving complex, sparse-reward tasks. The LARM effectively decomposes a sparsely rewarded problem into a sequence of sub-goals, providing a dense, structured reward signal that serves as a learning curriculum.

Figure 19 illustrates this process for the `UnlockToUnlock` task (see Appendix A.9 for the full RM). The plot shows that during training, the agent first learns to make incremental progress by maximizing the LARM reward (blue curve), which is awarded for completing key sub-goals like collecting keys and opening doors. Once the agent has reliably learned to follow this reward curriculum to its completion (indicated by the dashed line), the final task success rate (orange curve), which corresponds to a single sparse reward for reaching the goal, rises sharply. This demonstrates that the LARM successfully bridges the credit assignment gap, enabling the agent to solve a task that would otherwise be intractable due to the sparse environment reward.

#### A.7.3 MiniGrid - Longer Training

For completeness, we provide extended training results for the DoorKey experiments. Figure 20 shows the learning curves for the same agents when trained for 1M and 10M steps. These results confirm that performance saturates relatively early, validating our decision to focus on the initial phase of learning in the main paper's analysis.

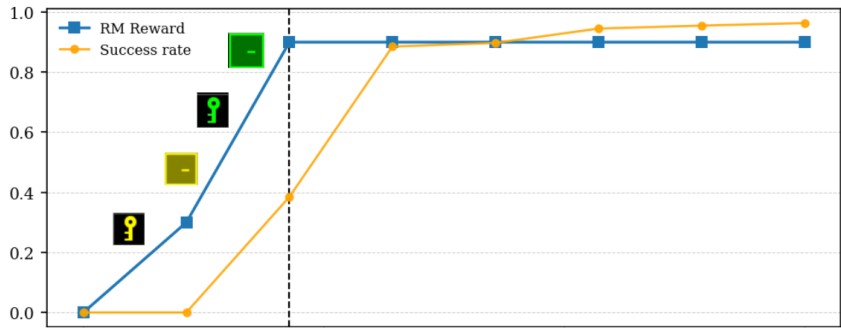

Figure 19: Analysis of LARM rewards during training in the `UnlockToUnlock` environment. The agent first learns to maximize the structured reward provided by the LARM for completing sub-goals (blue curve). Once the sub-goal sequence is mastered (dashed line), the agent rapidly achieves a high success rate on the sparsely-rewarded final objective (orange curve).

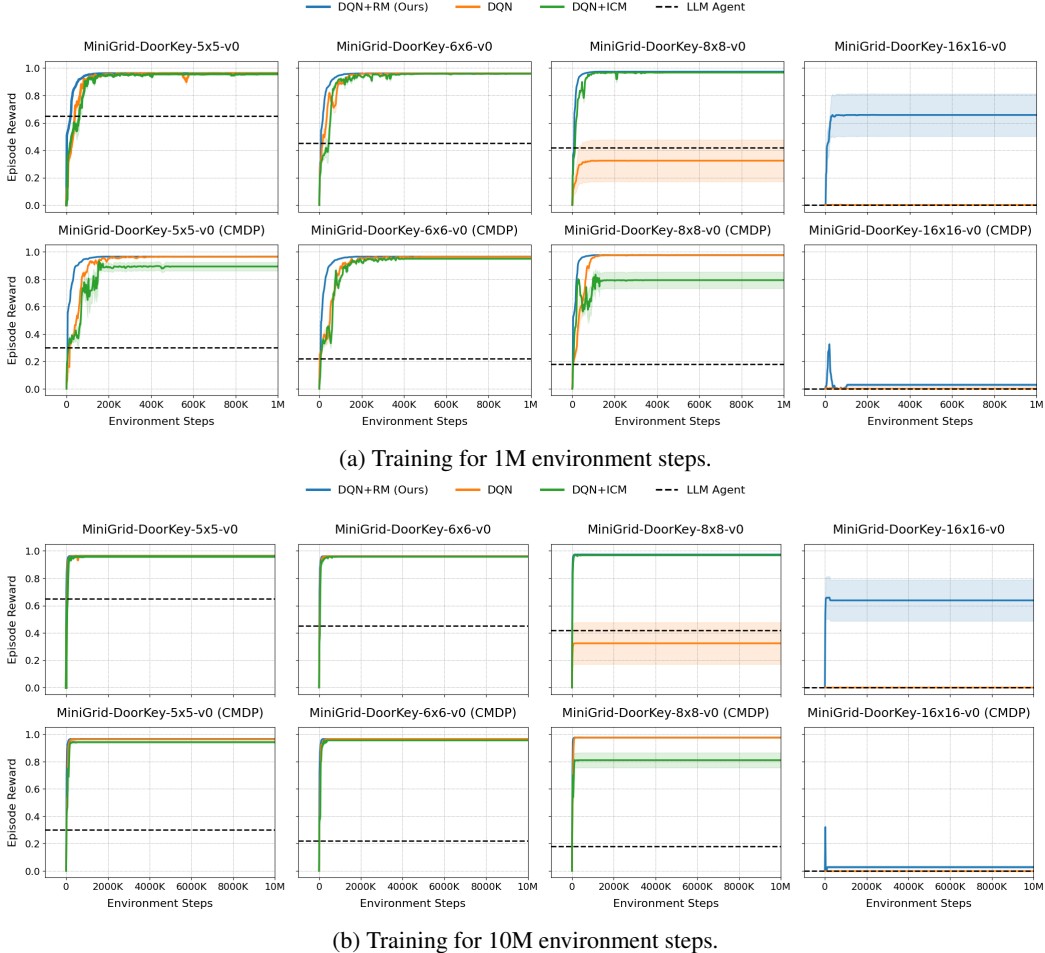

Figure 20: Extended training runs for the DoorKey experiments shown in Figure 6. The plots show performance up to 1M steps (a) and 10M steps (b). As agent performance plateaus early in training (around 300k steps), we present the shorter horizon in the main paper for clarity.

### A.7.4 MINIGRID - VFMS AS ZERO-SHOT REWARD MODELS

To provide a more comprehensive comparison, we evaluated the performance of a Vision-Language Model (VLM) used directly as a zero-shot reward function, following the methodology proposed by Rocamonde et al. (2023). For this baseline, we employed CLIP (Radford et al., 2021) to generate a dense reward signal. The reward at each timestep is calculated as the cosine similarity between the CLIP embedding of the current visual observation (an image of the environment state) and the embedding of a target language description specifying the task goal. We implemented the goal-baseline regularization technique from the original work to stabilize training, using a negative description as the baseline.

The positive goal descriptions and the shared baseline description for each MiniGrid task were specified as follows:

- **MiniGrid-DoorKey:** "The agent (red triangle) has opened the door (color-outlined square) and reached the goal room (green square)."

- **MiniGrid-BlockedUnlockPickUp:** "The agent (red triangle) has moved the ball (circle) away from the door (color-outlined square), has picked up the key, opened the door, and is now in the goal room (box square)."

- **MiniGrid-UnlockToUnlock:** "The agent (red triangle) has picked up both keys, opened both doors, and is now in the goal room (box square)."

- **Baseline (Negative Description):** "The agent (red triangle) is far from the goal, has not picked up any key, and has not opened any door."

The results of this baseline are presented in Figure 21. The CLIP-based reward model failed to make any progress across all evaluated tasks. Consequently, we omitted these results from the main paper for clarity. We hypothesize that this failure stems from known limitations of current VFMs, particularly their challenges with spatial reasoning and their struggle to interpret visually abstract or out-of-distribution environments like MiniGrid. As noted by Rocamonde et al. (2023), such failure modes are common when applying general-purpose VFMs to specialized domains that require nuanced visual understanding.

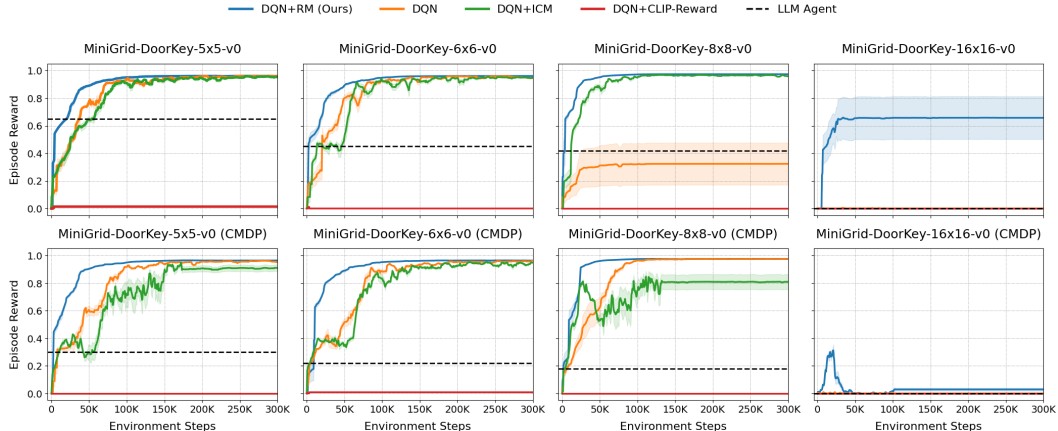

Figure 21: Performance of an agent trained using CLIP embeddings as a direct reward signal on MiniGrid. The VLM-based reward fails to provide a sufficient learning signal for the agent to make progress.

## A.8 PROMPTS

Below are the prompts for the generator and critic Foundation Models (FMs) for the DoorKey environment. This same prompt structure is used for all tasks, varying only the mission description for each environment and details on the specific environment API to generate the python labeling functions.

---

**Prompt: Reward Machine Generator**

**Environment:**

- **Agent:** A colored triangle.
- **Key:** Unlocks a door of the same color.
- **Door:** A color-outlined square within a wall.
- **Goal:** A colored square in a room (e.g., green).
- **Episode ends:** Upon reaching the goal (+1 reward) or reaching the step limit (0 reward).

**Mission:** *"This environment has a key that the agent must pick up in order to unlock a door and then get to the green goal square."*

---

**Your Role: Reward Machine Generator**
Generate a **concise**, **correct**, and **compact** reward machine in plaintext, wrapped in
```plaintext``` tags.
Your machine must:

1. **Densify the reward signal** to guide the agent effectively towards the goal.

2. **Use Boolean-predicate events** that are functions of the environment state. Do **not** use raw actions as events.

3. **Maximize compactness** with the fewest states and transitions possible, collapsing irrelevant events into a per-state (`state, else) -> state` transition.

4. **Adhere to the strict format** provided below. Do not add comments or extra text.

5. **Use clear event names** that are valid Python function names.

**Action Set (for reference):** `turn_left, turn_right, move_forward, pickup, drop, toggle`

```
REWARD_MACHINE:
STATES: u0, u1, ...
INITIAL_STATE: u0
TRANSITION_FUNCTION:
(u0, <event>) -> u1
(u0, else) -> u0
...
REWARD_FUNCTION:
(u0, <event>, u1) -> X
...
```

*Only list non-zero rewards in the REWARD_FUNCTION. All other transitions assume a reward of 0.*

Generate the reward machine now.

---

**Instructions: Reward Machine Critic**

**Your Role: Reward Machine Critic**
Evaluate a candidate reward machine for the MiniGrid environment. Focus on correctness, compactness, completeness, and format.

---

0. **Compactness**
    - Ensure the fewest possible states and transitions are used.
    - All irrelevant or zero-reward events in a state **must** be collapsed into a single `(state, else) -> state` transition.

1. **Boolean-Predicate Events**
    - Confirm that each transition's event is a Boolean predicate, **not** a raw action.
    - These predicates must reflect meaningful state conditions.

2. **Coverage of Events**
    - Every possible change in key predicates must be either explicitly handled or aggregated under that state's `else` transition.
    - Identify missing edge-case predicates

3. **Dense Rewards + Penalties**
    - Check for positive rewards on transitions that signify progress.
    - Verify that penalties or zero-rewards are used for regressions
    - Suggest additions for under-penalized failure modes.
    - Ensure reward magnitudes do not allow for reward hacking

4. **Mission Logic**
    - Ensure the sequence of states correctly enforces the logic required to solve the task.
    - Verify there are no unreachable states or unintentional loops.

5. **Format & Clarity**
    - The submission must strictly follow the specified format:
      ```
      REWARD_MACHINE:
      STATES: u0, u1, ...
      INITIAL_STATE: u0
      TRANSITION_FUNCTION:
      (u0, <event>) -> u1
      (u0, else) -> u0
      ...
      REWARD_FUNCTION:
      (u0, <event>, u1) -> X
      ...
      ```
    - **Only non-zero** rewards should be listed in the `REWARD_FUNCTION`.
    - There must be **no comments or extra text** within the plaintext block.
    - Event names must be descriptive Boolean predicates

---

**Your Response Format:**
- Cite specific transitions or sections of the machine in your evaluation.
- List concrete, actionable changes
- Be concise and to the point, while not missing any important details.

End your response with one of the following two verdicts:
- **NO CHANGES NEEDED**
- **CHANGES REQUIRED** followed by a bullet-list of the necessary fixes.

---

**Labeling Function Generator**

- **Task:** Implement each event from the reward machine below as a Python boolean function. Each function must return `True` if the event condition holds in the current state (`env`), otherwise `False`.
- **Reward machine:** `{REWARD_MACHINE}`
- **Guidelines:**
  - **Function Naming and Signatures**
    * Define one function per event in the RM.
    * Each function name must **exactly match** an event name.
    * Each function should take only `env` as its argument.
  - **Implementation Rules**
    * Use only the environment attributes and methods below:
      · `env.grid.get(i, j)` — Access object at (i, j)
      · `env.agent_pos` — Agent's position
      · `env.agent_dir` — Agent's direction (0-3)
      · `env.carrying` — Object agent is holding, e.g., a Key or `None`
      · `env.width`, `env.height` — Grid dimensions
  - **Object Information**
    * `WorldObj` is the base class.
      · A `Door` has `.is_open` and `.is_locked` attributes.
      · A `Key` has `.type == "key"`.
      · A `Goal` has `.type == "goal"`.
    * You cannot import classes. Instead, check object attributes (e.g., `obj.type == "key"`).
  - **Output Rules**
    * Only output clean, valid Python code.
    * No comments, explanations, or extra output.
    * Do not define a function for the `else` event.
    * Wrap your final output in triple backticks with a `python` tag for formatting.

---

## Labeling Functions Critic

**Your Role: Event-Function Critic**
You are evaluating Python functions that implement Boolean event predicates for a given reward machine (RM). Your job is to verify that the logic is correct, complete, and aligned with the RM specification.

**Task that the given RM should solve:**
*"This environment has a key that the agent must pick up in order to unlock a door and then get to the green goal square."*

---

## Evaluation Criteria

### 1. Boolean Predicate Fidelity

- Each function name must **exactly match** an event name from the RM.

- Each unique event in the RM must have a corresponding function.

- The function must return `True` if and only if the corresponding predicate becomes true in the current environment state.

### 2. Coverage & Scope

- Every event in the RM must have a corresponding function.

- There should be no extra functions that are not used in the RM.

**3. Correct Use of `env` API**  The following attributes and methods are available from the `env` object.

```
env.grid.get(i, j)     # Access object at (i, j)
env.agent_pos          # (x, y) position of the agent
env.agent_dir          # Integer: direction the agent is facing
env.carrying           # Object being carried (or None)
env.width, env.height  # Dimensions of the grid
```

Object types must be checked by attribute, as classes cannot be imported:

- A `Door` has `.is_open` and `.is_locked` attributes.

- A `Key` has `.type == "key"`.

- A `Goal` has `.type == "goal"`.

Functions must inspect these properties to determine predicate truth.

### 4. Clarity & Format

- Each function must be standalone, containing only executable code.

- Do not include comments, extra `print` statements, or surrounding explanations.

- The code should be clean, idiomatic Python.

---

## When You Respond

- Point out any **missing**, **misnamed**, or **extraneous** functions.

- Highlight any logic that is **incomplete**, **incorrect**, or **inefficient**.

- Suggest **precise code fixes** where needed.

- End your review with one of the following two verdicts, exactly as shown:

```
NO CHANGES NEEDED
```

or

```
CHANGES REQUIRED
- [List of bullet-pointed issues and suggested fixes]
```

## A.9 REWARD MACHINES

**DoorKey**

```
REWARD_MACHINE:
STATES: u0, u1, u2, u3
INITIAL_STATE: u0
TRANSITION_FUNCTION:
(u0, has_key) -> u1
(u0, else) -> u0
(u1, is_door_in_env_open) -> u2
(u1, not_has_key) -> u0
(u1, else) -> u1
(u2, at_goal) -> u3
(u2, else) -> u2
(u3, else) -> u3
REWARD_FUNCTION:
(u0, has_key, u1) -> 0.2
(u1, is_door_in_env_open, u2) -> 0.3
(u1, not_has_key, u0) -> -0.2
(u2, at_goal, u3) -> 1.0
```

**BlockedUnlockPickup**

```
REWARD_MACHINE:
STATES: u0, u1, u2, u3, u4
INITIAL_STATE: u0
TRANSITION_FUNCTION:
(u0, has_ball) -> u1
(u0, else) -> u0
(u1, has_key) -> u2
(u1, else) -> u1
(u2, door_unlocked) -> u3
(u2, no_key) -> u1
(u2, else) -> u2
(u3, has_box) -> u4
(u3, else) -> u3
(u4, else) -> u4
REWARD_FUNCTION:
(u0, has_ball, u1) -> 0.2
(u1, has_key, u2) -> 0.2
(u2, door_unlocked, u3) -> 0.2
(u2, no_key, u1) -> -0.3
(u3, has_box, u4) -> 1
```

**UnlockToUnlock**

```
REWARD_MACHINE:
STATES: u0, u1, u2, u3, u4, u5
INITIAL_STATE: u0

TRANSITION_FUNCTION:
(u0, got_y_key) -> u1
(u0, else) -> u0
(u1, door_y_opened) -> u2
(u1, lost_y_key) -> u0
(u1, else) -> u1
(u2, got_r_key) -> u3
(u2, else) -> u2
(u3, door_r_opened) -> u4
(u3, lost_r_key) -> u2
(u3, else) -> u3
(u4, entered_goal_room) -> u5
(u4, got_ball) -> u5
(u4, else) -> u4
(u5, else) -> u5

REWARD_FUNCTION:
(u0, got_y_key, u1) -> 0.1
(u1, door_y_opened, u2) -> 0.2
(u1, lost_y_key, u0) -> -0.1
(u2, got_r_key, u3) -> 0.1
(u3, door_r_opened, u4) -> 0.2
(u3, lost_r_key, u2) -> -0.1
(u4, entered_goal_room, u5) -> 0.3
(u4, got_ball, u5) -> 1
```

**KeyCorridor**

```
REWARD_MACHINE:
STATES: u0, u1, u2, u3, u4
INITIAL_STATE: u0
TRANSITION_FUNCTION:
(u0, on_purple_door_and_not_has_key) -> u1
(u0, else) -> u0
(u1, got_key) -> u2
(u1, else) -> u1
(u2, on_purple_door_and_has_key) -> u3
(u2, opened_red_door) -> u4
(u2, else) -> u2
(u3, opened_red_door) -> u4
(u3, else) -> u3
(u4, else) -> u4
REWARD_FUNCTION:
(u0, on_purple_door_and_not_has_key, u1) -> 0.1
(u1, got_key, u2) -> 0.2
(u2, on_purple_door_and_has_key, u3) -> 0.25
(u2, opened_red_door, u4) -> 0.5
(u3, opened_red_door, u4) -> 0.5
```

**Craftium**

```
REWARD_MACHINE:
STATES: u0, u1, u2, u3
INITIAL_STATE: u0
TRANSITION_FUNCTION:
(u0, get_wood) -> u1
(u0, else) -> u0
(u1, get_stone) -> u2
(u1, else) -> u1
(u2, get_iron) -> u3
(u2, else) -> u2
(u3, get_diamond) -> u4
(u3, else) -> u3
REWARD_FUNCTION:
(u0, get_wood, u1) -> 0.25
(u0, get_stone, u1) -> 0.5
(u0, get_iron, u1) -> 0.75
(u0, get_diamond, u1) -> 1.25
```

**Metaworld**

```
REWARD_MACHINE:
STATES: u0, u1, u2, u3, u4
INITIAL_STATE: u0
TRANSITION_FUNCTION:
(u0, near_object) -> u1
(u0, grasp_success) -> u2
(u0, else) -> u0
(u1, grasp_success) -> u2
(u1, not_near_object) -> u0
(u1, else) -> u1
(u2, not_grasp_success) -> u0
(u2, object_near_goal) -> u3
(u2, success) -> u4
(u2, else) -> u2
(u3, not_object_near_goal) -> u2
(u3, success) -> u4
(u3, else) -> u3
(u4, else) -> u4
REWARD_FUNCTION:
(u0, near_object, u1) -> 0.20
(u1, grasp_success, u2) -> 0.40
(u0, grasp_success, u2) -> 0.40
(u1, not_near_object, u0) -> -0.20
(u2, not_grasp_success, u0) -> -0.40
(u2, object_near_goal, u3) -> 0.80
(u3, not_object_near_goal, u2) -> -0.80
(u2, success, u4) -> 1.50
(u3, success, u4) -> 1.50
```

### A.10 LABELING FUNCTIONS

**Labeling Functions for DoorKey**

```python
def has_key(env):
    return env.carrying is not None and env.carrying.type == "key"

def is_door_in_env_open(env):
    for i in range(env.height):
        for j in range(env.width):
            obj = env.grid.get(j, i)
            if obj is not None and obj.type == "door" and obj.
                is_open:
                return True
    return False

def not_has_key(env):
    return not (env.carrying is not None and env.carrying.type == "
        key")

def at_goal(env):
    x, y = env.agent_pos
    obj = env.grid.get(x, y)
    return obj is not None and obj.type == "goal"
```

**Labeling Functions for BlockedUnlockPickup**

```python
def has_ball(env):
    return env.carrying is not None and env.carrying.type == "ball"

def has_key(env):
    return env.carrying is not None and env.carrying.type == "key"

def door_unlocked(env):
    for i in range(env.width):
        for j in range(env.height):
            obj = env.grid.get(i, j)
            if obj is not None and hasattr(obj, "is_locked") and
                obj.is_locked == False:
                return True
    return False

def no_key(env):
    return not has_key(env)

def has_box(env):
    return env.carrying is not None and env.carrying.type == "box"
```

**Labeling Functions for UnlockToUnlock**

```python
def got_y_key(env):
    return (
        env.carrying is not None
        and getattr(env.carrying, "type", None) == "key"
        and getattr(env.carrying, "color", None) == "yellow"
    )

def door_y_opened(env):
    for i in range(env.width):
        for j in range(env.height):
            obj = env.grid.get(i, j)
            if (
                obj is not None
                and getattr(obj, "type", None) == "door"
                and getattr(obj, "color", None) == "yellow"
                and getattr(obj, "is_open", False)
            ):
                return True
    return False

def lost_y_key(env):
    return not (
        env.carrying is not None
        and getattr(env.carrying, "type", None) == "key"
        and getattr(env.carrying, "color", None) == "yellow"
    )

def got_r_key(env):
    return (
        env.carrying is not None
        and getattr(env.carrying, "type", None) == "key"
        and getattr(env.carrying, "color", None) == "red"
    )

def door_r_opened(env):
    for i in range(env.width):
        for j in range(env.height):
            obj = env.grid.get(i, j)
            if (
                obj is not None
                and getattr(obj, "type", None) == "door"
                and getattr(obj, "color", None) == "red"
                and getattr(obj, "is_open", False)
            ):
                return True
    return False

def lost_r_key(env):
    return not (
        env.carrying is not None
        and getattr(env.carrying, "type", None) == "key"
        and getattr(env.carrying, "color", None) == "red"
    )

def entered_goal_room(env):
    # Example check: agent is in the leftmost 5 columns.
    return env.agent_pos[0] < 5

def got_ball(env):
    return (
        env.carrying is not None
        and getattr(env.carrying, "type", None) == "ball"
    )
```

**Labeling Functions for KeyCorridor**

```python
def on_purple_door_and_not_has_key(env):
    i, j = env.agent_pos
    obj = env.grid.get(i, j)
    if obj is not None and hasattr(obj, 'type') and obj.type == '
        door' and getattr(obj, 'color', None) == 'purple':
        if env.carrying is None or (hasattr(env.carrying, 'type')
             and env.carrying.type != 'key'):
            return True
    return False

def got_key(env):
    return (
        env.carrying is not None
        and hasattr(env.carrying, 'type')
        and env.carrying.type == 'key'
    )

def on_purple_door_and_has_key(env):
    i, j = env.agent_pos
    obj = env.grid.get(i, j)
    if obj is not None and hasattr(obj, 'type') and obj.type == '
        door' and getattr(obj, 'color', None) == 'purple':
        if env.carrying is not None and hasattr(env.carrying, 'type
            ') and env.carrying.type == 'key':
            return True
    return False

def opened_red_door(env):
    i, j = env.agent_pos
    obj = env.grid.get(i, j)
    if obj is not None and hasattr(obj, 'type') and obj.type == '
        door' and getattr(obj, 'color', None) == 'red':
        if getattr(obj, 'is_open', False) is True:
            return True
    return False

def reached_goal(env):
    i, j = env.agent_pos
    obj = env.grid.get(i, j)
    if obj is not None and hasattr(obj, 'type') and obj.type == '
        goal':
        return True
    return False
```

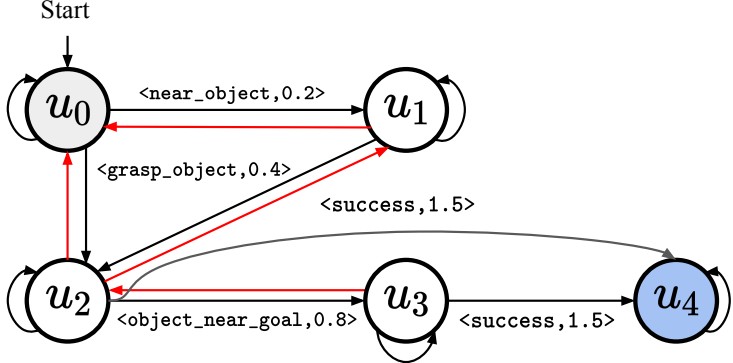

Figure 22: A visualization of the Meta-World reward machine. Red arrows indicate a negative reward when the state transitions in the opposite direction, moving further away from the success state $u_4$. For example, if the agent is in state $u_1$ and transitions back to state $u_0$, it will receive a reward of $-0.2$. When the agent does not trigger any transition event, the state remains the same, as indicated by the self-loop arrows.

### A.10.1 A VISUALIZATION OF META-WORLD REWARD MACHINE

Figure 22 shows a visualization of the Meta-World reward machine generated by the FM. Red arrows indicate a reversed path in which the reward machine's state transitions further away from the success state.

## A.11 HYPERPARAMETERS

### A.11.1 DQN (MINIGRID & BABYAI)

The hyperparameters listed in Table 5 were used for all DQN, DQN+RND, and DQN+RM agents in the MiniGrid and BabyAI environments.

Table 5: DQN hyperparameters used for all MiniGrid and BabyAI experiments.

| Hyperparameter | Value |
|---|---|
| Total Timesteps | $1 \times 10^7$ |
| Learning Rate | $1 \times 10^{-4}$ |
| Replay Buffer Size | $1 \times 10^6$ |
| Learning Starts | $80,000$ |
| Batch Size | $32$ |
| Discount Factor ($\gamma$) | $0.99$ |
| Target Network Update Frequency | $2,500$ |
| Target Network Update Rate ($\tau$) | $1.0$ |
| Train Frequency | $4$ |
| *Epsilon-Greedy Exploration* | |
| Initial Epsilon ($\epsilon_{start}$) | $1.0$ |
| Final Epsilon ($\epsilon_{end}$) | $0.01$ |
| Exploration Fraction | $0.35$ |
| Double Q-Learning | False |

Table 6: PPO hyperparameters used for the Craftium experiments.

| Hyperparameter | Value |
|---|---|
| Total Timesteps | $1 \times 10^7$ |
| Number of Parallel Environments | 4 |
| Steps per Environment (Rollout) | 128 |
| Number of Minibatches | 4 |
| PPO Update Epochs | 4 |
| *Optimizer and Learning Rate* | |
| Learning Rate | $5 \times 10^{-5}$ |
| Learning Rate Annealing | True |
| Max Gradient Norm | 0.5 |
| *PPO & GAE Parameters* | |
| Discount Factor ($\gamma$) | 0.99 |
| GAE Lambda ($\lambda$) | 0.95 |
| Clipping Coefficient | 0.1 |
| Value Function Loss Clipping | True |
| Advantages Normalization | True |
| *Loss Coefficients* | |
| Entropy Coefficient | 0.01 |
| Value Function Coefficient | 0.5 |

### A.11.2 PPO (CRAFTIUM)

For the more computationally demanding Craftium environment, we use PPO to leverage vectorized rollouts for faster training. The hyperparameters for the PPO agent, which were kept consistent for both the baseline and our method, are detailed in Table 6.

### A.11.3 RAINBOW (XLAND-MINIGRID)

For the experiments in XLand-MiniGrid, we use a Rainbow DQN agent. The hyperparameters, consistent for both the baseline and our method, are detailed in Table 7.

### A.11.4 SAC (META-WORLD)

In Meta-World experiments we use SAC (Haarnoja et al., 2018) to train the sparse reward and the agent augmented with the reward machine. The hyperparameters are detalied in Table 8

Table 7: Rainbow DQN hyperparameters used for the XLand-MiniGrid experiments.

| Hyperparameter | Value |
|---|---|
| *Base DQN Parameters* | |
| Total Timesteps | $5 \times 10^6$ |
| Learning Rate | $6.25 \times 10^{-5}$ |
| Replay Buffer Size | $1 \times 10^6$ |
| Learning Starts | $80,000$ |
| Batch Size | $32$ |
| Discount Factor ($\gamma$) | $0.99$ |
| Target Network Update Frequency | $5,000$ |
| Train Frequency | $4$ |
| *Epsilon-Greedy Exploration* | |
| Initial Epsilon ($\epsilon_{start}$) | $1.0$ |
| Final Epsilon ($\epsilon_{end}$) | $0.05$ |
| Exploration Fraction | $0.1$ |
| *Rainbow Components* | |
| N-step Learning | $3$ |
| PER Alpha ($\alpha$) | $0.5$ |
| PER Initial Beta ($\beta_0$) | $0.4$ |
| Distributional Atoms | $51$ |
| Distributional Value Range ($V_{min}, V_{max}$) | $[-10, 10]$ |

Table 8: SAC hyperparameters used for the Meta-World experiments.

| Hyperparameter | Value |
|---|---|
| *Base SAC Parameters* | |
| Total Timesteps | $1.5 \times 10^7$ |
| Learning Rate | $3 \times 10^{-4}$ |
| Replay Buffer Size | $1 \times 10^6$ |
| Learning Starts | $5,000$ |
| Batch Size | $512$ |
| Discount Factor ($\gamma$) | $0.99$ |
| Target Network Update coefficient | $0.005$ |
| Policy Train Frequency | $2$ |
| Critic Train Frequency | $1$ |
| *Exploration* | |
| Intrinsic Reward model | RND |
| Intrinsic Reward Coefficient | $0.01$ |

