# OpenReview forum: "ARM-FM: Automated Reward Machines via Foundation Models for Compositional Reinforcement Learning"
_ICLR.cc/2026/Conference — ICLR 2026 Poster_

### Official Review · Reviewer_tffC · 2025-10-27

**Soundness:** 3
**Presentation:** 3
**Contribution:** 3
**Rating:** 6
**Confidence:** 3

**Summary:**

This paper presents a framework using foundation models (FMs) to automate Reward Machine (RM) generation from natural language specifications for reinforcement learning (RL). The core idea is to leverage FMs to produce not only the RM's automata structure but also the executable labeling functions to address sparse-reward, long-horizon embodied control tasks. The framework also proposes conditioning the RL policy on language embeddings of RM states to enable compositional generalization. The proposed method demonstrates its effectiveness by solving several challenging benchmark tasks (MiniGrid, Craftium, MetaWorld) intractable for standard RL baselines.

**Strengths:**

1. Effective FMs and RM pipeline with strong empirical results on sparse, long-horizon RL benchmarks (MiniGrid, Craftium, MetaWorld) where baselines (ICM, PPO, SAC) fail.
2. By generating an explicit RM automaton, the framework offers a degree of interpretability and debuggability compared to opaque, end-to-end FM reward signals.
3. Robust automated labeling code gen through LLM generator-critic loop for plan grounding.
4. The use of language embeddings for the policy conditioning enables zero-shot generalization on a compositional task.

**Weaknesses:**

1. The grounding mechanism (using labeling functions) relies on access to the simulator's high-level, symbolic state representation (env.carrying, obj.type, obj.is_open). This might limit the framework's direct applicability to real-world scenarios or learning from raw pixels/sensory data, a common limitation but still noteworthy.

2. While LLMs-driven RM generation and labeling functions shows novelty in automation, the strong performance on non-Markovian tasks feels incremental, as RMs' efficacy in such settings is well-established (e.g., [1, 2, 3]). Expected results limit surprise, though the proposed framework remains valuable for practitioners.

[1] Rodrigo Toro Icarte, Toryn Q Klassen, Richard Valenzano, and Sheila A McIlraith. Reward machines:
Exploiting reward function structure in reinforcement learning. Journal of Artificial Intelligence
Research, 73:173–208, 2022.

[2] Rodrigo Toro Icarte, Toryn Klassen, Richard Valenzano, and Sheila McIlraith. Using reward machines
for high-level task specification and decomposition in reinforcement learning. In International
Conference on Machine Learning, pp. 2107–2116. PMLR, 2018.

[3] Andrew Li, Zizhao Chen, Toryn Klassen, Pashootan Vaezipoor, Rodrigo Toro Icarte, and Sheila
McIlraith. Reward machines for deep rl in noisy and uncertain environments. Advances in Neural
Information Processing Systems, 37:110341–110368, 2024.

**Questions:**

1. How sensitive is the framework to errors or inefficiencies in the FM-generated labeling functions, especially for more complex environments? What is the typical failure rate, and how much manual correction was needed during experiments?

2. What are the concrete next steps or necessary innovations required to extend ARM-FM to environments where only raw observations (e.g., pixels) are available, requiring the labeling function predicates themselves to be learned?

3. While RMs offer structure, manually designing them becomes intractable for complex tasks. Does the FM-based generation approach scale effectively? Were there task complexities or types where the FM struggled significantly to produce a useful RM structure, even with the critic loop?

4. Regarding the zero-shot generalization results (e.x., Figure 10), success relies on the new task's sub-goal embeddings being 'close to an embedding seen during training.' How can we quantify this 'closeness' in the embedding space? For example, does a specific cosine similarity threshold between a new sub-goal embedding and the closest training sub-goal embedding correlate with successful transfer?

---

> ### Author Response · Authors · 2025-11-20
>
> We thank the reviewer for the feedback and appreciate their highlighting of our “effective FMs and RM pipeline,” the interpretability of generating an explicit automaton, the “robust automated labeling… via the generator-critic loop,” and the zero-shot generalization enabled by “language embeddings for the policy”. Below, we respond to the reviewer’s feedback. **We also encourage the reviewer to read our general comment and the revised manuscript, where all changes are highlighted in blue.**
>
> **Q1) Reliance on symbolic simulator state may limit real-world applicability.**
>
> We agree that for the results reported in this paper, using code-based labeling functions that access simulator state bypasses this important problem. This was a deliberate choice to isolate and evaluate the core LARM generation component, separate from the challenges of perception. However, we argue that this does not limit the ARM-FM framework itself. Our formalization in Section 2 is general and admits any set of boolean predicates for the labeling functions. This means they could be learned logical predicates or, as we discuss in our response to **Q4**, perception-based modules or queries to another FM. We elaborate more on this topic in our answer to **Q4**.
>
> **Q2) Performance on non-Markovian tasks feels incremental.**
>
> We agree that RMs are already known to work well in non-Markovian settings: **this is exactly why we use them as the structural backbone for FM-generated task knowledge**.
>
> Our contribution is not showing that RMs work, but making them scalable and automatable.  Our step is non-incremental in three ways:
>
> - Scale: We auto-generate and solve 1,000+ LARMs in XLand-MiniGrid, far beyond what manually designed or simpler RM methods could handle.
>
> - Breadth: The framework works across diverse domains (MiniGrid, Craftium, Meta-World) without special-case engineering.
>
> - Semantic Generalization: We introduce FM-generated semantic embeddings for LARM states. This grounding enables strong zero-shot generalization (Sec. 3.4).
>
> **Q3) Sensitivity to FM errors in labeling functions.**
>
> Empirically, FM-generated labeling functions were robust: they were correct for all Craftium tasks and all 1,000+ XLand-MiniGrid LARMs. Only 3 tasks required manual correction. In each case, the fix involved providing high-level natural-language feedback (e.g., noting a missed edge case), which the generator-critic loop incorporated. This transparency and seamless human input are practical strengths of the framework.
>
> We now provide more details in Appendix A.4.
>
> **Q4) Next steps for raw-pixel environments where predicates must be learned.**
>
> Our work used code-based labelling functions accessing the simulator state to isolate and evaluate the FM's ability to generate the compositional task structure (the LARM) and its embeddings. However, the ARM-FM framework is designed to be general, and the concrete next step is to ground these labelling functions in raw observations. Our framework's separation of high-level logic (LARM) from low-level event detection makes this a 'pluggable' problem. Future work might replace our code-based predicates with perception-based ones. For instance, labelling functions could be:
>
> - A VLM query, where the LARM state (e.g., 'get the key') provides semantic context for the query (e.g., 'Is the agent holding the key in this image?').
> - A learned predicate, where the FM generates the target concept (e.g., has_key), and a small, dedicated perception module is trained from pixels to detect that specific event.
>
> This flexibility is why we have now added the clarification in Section 2.1: 'While we use code to define labeling functions in this work, the ARM-FM framework is general, supporting any boolean predicate (e.g. formal logic, or queries to other FMs).
>
> **Q5) Does FM-based RM generation scale, and where does it struggle?**
>
> Yes. Scalability is one of ARM-FM’s main strengths: the system generated 1,000+ LARMs for XLand-MiniGrid, which would be impossible to author manually. We have now documented failures in Appendix A.4 (Table 3). The FM struggled mainly with: (1) logical edge cases (e.g., dropping a key after picking it up), and (2) overly sparse logic, which makes RL too hard. These are classic challenges, and in each case, a small natural-language correction fixed the issue. Figure 11 (left) also shows that failure rates decrease with model scale, suggesting strong scalability.
>
> **Q6) Quantifying “closeness” for zero-shot generalization.**
>
> Our hypothesis is that embedding proximity enables transfer. Figure 11 (right) provides evidence: semantically similar sub-goals cluster tightly in the embedding space, indicating that cosine similarity meaningfully tracks task semantics. This structured embedding geometry is what allows the policy to generalize to unseen sub-goals that embed near ones encountered during training.

---

> > ### Comment · Reviewer_tffC · 2025-11-24
> >
> > Thank you to the authors for the detailed and thoughtful rebuttal. The responses adequately address my concerns, and the clarifications are helpful. I will keep my original score.

---

> > > ### Author Response · Authors · 2025-11-26
> > >
> > > We thank the reviewer for considering our rebuttal. Are there any remaining concerns we could address that would make you more supportive of our submission?

---

> > > > ### Comment · Reviewer_tffC · 2025-11-26
> > > >
> > > > I truly appreciate the effort you put into the clarifications. However, I would like to keep my current score since the study is not reproducible as the code is not publicly available. Also, I still think that this line of research is not breakthrough research, instead, smartly, implement to apply well known paradigm to scale up even though I agree that this is also a good contribution.
> > > > I sincerely wish you the best with the review process.

---

### Official Review · Reviewer_3qmz · 2025-10-29

**Soundness:** 3
**Presentation:** 4
**Contribution:** 4
**Rating:** 8
**Confidence:** 4

**Summary:**

The paper presents a novel methodology for tackling complex, long-horizon tasks in reinforcement learning (RL). At the core of the approach is the use of a foundational model (FM) to produce a high-level task decomposition, which is formalized as a Language-Aligned Reward Machine (LARM). In this framework, each state of the reward machine is annotated with a natural language description of the corresponding subgoal, enabling the agent to condition its policy on these instructions. This alignment facilitates goal-directed behavior and supports zero-shot generalization to previously unseen tasks.

As the agent transitions through the reward machine states, it receives structured intermediate rewards that guide learning and enhance sample efficiency. Notably, the FM also produces executable code to define the labeling function, which determines when a subtask has been completed. In doing so, the FM provides all essential components—task decomposition, subgoal descriptions, intermediate rewards, and labeling logic—allowing the agent to leverage LARM in novel environments with minimal human supervision.

Experimental results across a diverse set of benchmarks demonstrate the effectiveness of the proposed approach, outperforming strong baselines such as RL with intrinsic motivation and LLM agents.

**Strengths:**

I thoroughly enjoyed reading this paper. It presents a compelling case for the use of foundational models (FMs) and reward machines (RMs) as effective tools for communicating task structure to reinforcement learning (RL) agents. Particularly noteworthy is the FM’s ability to autonomously generate all necessary components—from the RM structure to the labeling function—based solely on a natural language task description and an image of the environment. The idea of associating natural language instructions with each RM state is especially interesting, as it enables zero-shot generalization and, depending on the quality of the subgoal descriptions, can even lead to near-optimal task execution despite the decomposition into potentially myopic subtasks.

The paper also offers a broad and thorough empirical evaluation, spanning environments from 2D grid worlds to complex 3D settings in robotics and Minecraft. Across these diverse domains, the proposed method consistently demonstrates strong performance. The authors further validate their approach by testing generalization capabilities and conducting a sensitivity analysis, highlighting the importance of both intermediate rewards and state embeddings to the agent’s success.

Finally, I believe this work opens up several promising avenues for future research, some of which I have outlined in the weaknesses section of my review.

**Weaknesses:**

While I found the paper insightful and well-executed overall, there are some areas where further clarification or experimentation would strengthen its contributions.

First, the role of human verification in Figure 3 is underexplored. The only mention of this “human-in-the-loop” process appears in the final paragraph of the conclusion. From what I understand, the human reviewer inspects the LARM generated by the FM and may reject it. However, it remains unclear what happens after a rejection: Does the human refine the prompt, or simply initiate additional rounds of self-improvement? Moreover, how frequently were LARMs rejected or refined during the experiments? What were the typical reasons for rejection? For instance, were there cases where bugs in the labeling function or poorly designed intermediate rewards prevented task completion? Clarifying whether such issues arose—and how they were handled—would help contextualize the robustness of the method. Alternatively, if no human verification was performed during experiments, that should be explicitly stated.

Second, in Section 3.1, the paper attributes the superior performance of DQN+RM primarily to the intermediate rewards provided by the reward machine. While this is a plausible explanation, I believe the state embeddings may also play a significant role. Since the environments are partially observable, the RM state effectively acts as a form of memory, allowing the agent to retain contextual information across time steps. In contrast, the baseline agents are memoryless, which may explain their underperformance—not necessarily due to the absence of intermediate rewards, but because they lack the capacity to maintain relevant state information. To better isolate the contribution of intermediate rewards, it would have been helpful to include the “No RM rewards” ablation from Section 3.4 in Figures 6 and 7. Additionally, comparing against an LSTM-based agent would help determine whether memory alone accounts for the observed performance gap. The same considerations apply to the experiments discussed in Section 3.2.

Third, while the zero-shot generalization result is promising, it currently feels somewhat anecdotal. The paper presents a single instance in which the agent successfully generalizes to a novel task. A more systematic evaluation—spanning a broader range of tasks—would strengthen the case for the generalization capabilities of LARMs.

Finally, a discussion on optimality would be valuable. The FM can describe RM states using either myopic or non-myopic natural language instructions. For example, a myopic instruction like “Get the key” may lead the agent to complete subtasks sequentially without considering long-term consequences, potentially resulting in suboptimal behavior (e.g., choosing a key that doesn’t lead to the door). In contrast, a non-myopic instruction such as “Get the key and then go to the door” could guide the agent toward more globally optimal behavior. It would be interesting to explore whether prompting the FM to produce non-myopic subgoal descriptions could improve long-term performance.

**Questions:**

1. Could you elaborate on the specific role and responsibilities of the human verifier during the experimental evaluation?
2. How frequently was the LARM generated by the FM deemed unsatisfactory by the human verifier?
3. What were the most common issues or failure modes observed in the LARM generation process by the FM?
4. When an LARM was rejected by the human verifier, what steps were taken next? Was the prompt revised or refined, and if so, how was this process carried out?
5. Does the labeling function rely solely on information observable by the agent, or does it access privileged information from the environment's internal state? If it uses privileged information, this could pose challenges for real-world deployment—particularly in sim-to-real scenarios—since such information may not be available outside the simulation.

---

> ### Author Response · Authors · 2025-11-20
>
> We thank the reviewer for the very positive feedback and are glad they “thoroughly enjoyed reading this paper” and found our use of FMs and RMs “a compelling case” for communicating structure to RL agents. We also appreciate their remarks on our “broad and thorough empirical evaluation” and their observation that associating language with RM states enables “zero-shot generalization.” Below, we respond to the reviewer’s feedback. **We also encourage the reviewer to read our general comment and the revised manuscript, where all changes are highlighted in blue.**
>
> **Q1) Could you elaborate on the specific role and responsibilities of the human verifier during the experimental evaluation? How frequently was the LARM generated by the FM deemed unsatisfactory by the human verifier? What were the most common issues or failure modes observed in the LARM generation process by the FM?**
>
> Thank you for raising this point. We've now clarified this in Appendix A.4.
>
> The human's role is not just a passive verifier; they can actively replace the FM critic during a self-improvement round. We built an interactive interface for this, allowing the human to provide natural language refinement (e.g., 'You missed the edge case where the agent drops the key') which is fed back to the generator.
>
> We see this as a feature of the LARM framework. It creates a human-interpretable channel to inspect and refine the reward design, which is crucial for building trust and handling complex edge cases.
>
> With regards to frequency, the fully automated loop was sufficient for the vast majority of our tasks. Human intervention was only needed for 3 specific tasks. The rest, including all 1,000+ LARMs for XLand-MiniGrid, were generated and verified fully automatically (using our LLM-as-judge method).
>
>
> **Q2) When an LARM was rejected by the human verifier, what steps were taken next? Was the prompt revised or refined, and if so, how was this process carried out?**
>
> As detailed in our new Appendix A.4, the process is interactive. The human doesn't just revise the initial prompt; they replace the FM critic for that refinement round. They see the FM's failed LARM and the FM critic's feedback, and then provide their own natural language comment.
>
> For example, for the 'KeyCorridor' task, the human's feedback was: 'The LARM is too sparse. Define intermediate rewards, perhaps for crossing doors or entering new rooms.' This comment is then fed back to the generator, which attempts a new LARM. This allows the human to provide high-level, semantic guidance without needing to write code or formal specs.
>
> **Q3) Does the labelling function rely solely on information observable by the agent, or does it access privileged information from the environment's internal state? If it uses privileged information, this could pose challenges for real-world deployment, particularly in sim-to-real scenarios, since such information may not be available outside the simulation.**
>
> This is a great point and a critical question for any real-world deployment. The reviewer is correct: in our simulated experiments (MiniGrid, MetaWorld), our labeling functions are implemented as boolean code functions that do access the underlying simulator state (e.g., agent/object positions, agent.has_key). We made this choice to isolate and evaluate the core contribution of our work: the FM's ability to generate the compositional reward structure (the LARM) and its embeddings, separate from the  problem of perception. However, this is not a fundamental limitation of the ARM-FM framework. The framework is flexible and only requires the labeling function to be any boolean evaluator. For a real-world system, we would simply replace these privileged-state functions with ones that operate on sensor data. For example, a labeling function like has_key() could be:
>
> - A VLM queried to recognize if the agent's gripper is holding a key from an image. As we note, this could be slow, but recent progress in fast VLMs makes this increasingly practical.
>
> - A simple, trained classifier: A small, dedicated perception model trained just to detect the specific 'key-in-hand' event.The key is that ARM-FM separates the high-level task logic (RM) from the low-level event detection (labeling functions).
>
> Our work proves the FM is highly effective at the former, and the framework's flexibility allows any 'real-world' perception module to be plugged in for the latter.

---

### Official Review · Reviewer_4WaP · 2025-11-01

**Soundness:** 3
**Presentation:** 3
**Contribution:** 2
**Rating:** 4
**Confidence:** 4

**Summary:**

The paper introduces ARM-FM, which uses a large FM to automatically generate RMs from natural language task descriptions. The generated LARM includes state-level natural language descriptions and labeling functions. Policies are trained on the product MDP of the environment and RM, conditioned on the embedding of the current RM state instruction. Experiments on MiniGrid, Craftium (3D), and other benchmarks demonstrate improved zero-shot generalization.

**Strengths:**

+ I believe the main strength of the paper is the use of language embeddings of RM state descriptions for conditioning the policy.
+ Using FMs to automate RM construction from natural language specifications is useful since manual RM design requires expertise.
+ The experiments and visualizations across multiple environments are commendable.

**Weaknesses:**

- My main concern is the limited novelty. RL over RMs and product MDPs is well established. The main addition here is automating RM generation with an FM and embedding the RM states, which is conceptually incremental in my view for ICLR.
- Another concern I have is that relation to prior work is not sufficiently clear. The distinction needs to be well articulated. This includes a large body of work on FM-guided RL frameworks such as ReAct, SayCan, Voyager, Eureka etc and the list goes on.
- Most tasks considered seem to involve small domain variation rather than truly different domains. It remains unclear whether the learned embeddings generalize under larger domain or modality gaps. So is the paper over claiming generalization?
- There are missing comparisons in evaluation with recent vision language or FM-guided RL frameworks
- The human verification weakens the claim of full automation.
- Reproducibility may be an issue.

**Questions:**

1) Precisely what is novel beyond FM-generated RM structure/code and conditioning on language embeddings of RM states? What core technical insight would not be obvious from prior RM + product-MDP and FM-guided RL work?
2) Can you provide a comparison table clarifying how your pipeline differs from closest work that already uses LMs to synthesize automata/RMs/LTL specs, and from FM-guided RL?
3) Is there a theoretical claim you can formalize?
4) What fraction of tasks required human verification/edits?
5) How are RM-state language embeddings obtained?
6) How large are the domain gaps actually used (measured by a concrete metric)? Can you report performance as a function of controlled gap magnitude and type?
7) Can you compare against stronger FM-guided baselines?

---

> ### Author Response · Authors · 2025-11-20
>
> We thank the reviewer for the feedback and are glad they highlighted “the use of language embeddings… for conditioning the policy” as the main strength, as well as the usefulness of “using FMs to automate RM construction” and the “commendable” experimental results. Below, we respond to the reviewer’s feedback. **We also encourage the reviewer to read our general comment and the revised manuscript, where all changes are highlighted in blue.**
>
> **Q1) What is novel beyond FM-generated RM structure/code and language-conditioned embeddings?**
>
> Our contribution is not a new RM formalism, but a practical pipeline that makes RMs usable at scale in high-dimensional, image-based, long-horizon RL settings (where prior RM methods operate only in small, symbolic, tabular domains).
>
> First, prior RM work assumes low-dimensional states and manually labeled predicates. In contrast, integrating RMs with visual observations and continuous control requires generating both symbolic conditions and product-MDP transitions (something previous work does not address). Our FM generator-critic loop produces these grounding functions automatically.
>
> Second, while the RMs remain compact, the process is fully automated grounding in large environments without hand-engineered predicates or manual transition definitions.
>
> Finally, conditioning the policy on semantic language embeddings is a key novelty: it provides a meaningful scaffold that enables zero-shot and compositional generalization, which prior RM and FM-RL work does not offer.
>
> **Q2) Relation to prior work (ReAct, SayCan, Voyager, Eureka, LLM, etc.)**
>
> We expanded our discussion of related work by adding a new subsection (Appendix A.6) and a detailed comparison table (Table 4). This table contrasts ARM-FM with both (i) FM-driven automata synthesis methods (e.g., L*LM, RAD, SAT-based RM learning), and (ii) FM-guided RL frameworks (ReAct, SayCan, Voyager, Eureka, Motif, ELLM). The comparison highlights differences in required supervision (e.g., demonstrations vs. none), how FMs are used (e.g., code generation, affordance scoring, reward evolution), assumptions about skills or intermediate abstractions, and whether the method produces an explicit RM structure. We also emphasize ARM-FM’s unique combination of end-to-end RM generation, semantic grounding, and learnable multi-task control. These additions are included in the revised manuscript (Appendix A.6).
>
> **Q3) Missing comparisons with stronger FM-guided baselines.**
>
> Our “LLM Agent” baseline is in fact ReAct, implemented via the BALROG benchmark. We chose ReAct because it is a strong, widely used FM-guided agent relying on in-context reasoning. ARM-FM significantly outperforms it.
>
> Other FM-guided methods were unsuitable due to domain specificity or computational constraints (Motif requires text captions; ELLM needs per-step VLM calls; Eureka requires access to the source code of the reward function; Voyager and SayCan depend on predefined skills). Thus ReAct is the most directly applicable baseline.
>
> We updated Section 3.1, Related Work, and Figures 6 and 7 to explicitly label this baseline as ReAct.
>
> **Q4) Are the generalization claims overstated?**
>
> To clarify, we distinguish two forms of generalization:
>
> In-distribution embedding generalization (Sec. 3.4): As the reviewer notes, this is within the combinatorial domain of XLand-MiniGrid. We do not claim the embeddings transfer across domains (e.g., to Meta-World).
>
> Cross-domain framework generalization: The method generalizes across very different modalities: MiniGrid (2D symbolic navigation), Craftium (3D voxel world), Meta-World (continuous control). The same ARM-FM pipeline applies without modification to all of these domains
>
> We updated the text to make this distinction explicit.
>
> **Q5) Is there a theoretical claim you can formalize?**
>
> Yes. Appendix A.5 now includes Proposition 1 (Optimality Preservation): if the LARM has no positive-reward cycles and the goal reward exceeds any sum of intermediate rewards, then any optimal policy for the dense LARM objective is also optimal for the original sparse MDP.
>
> This formalizes the idea that FM-generated intermediate rewards provide guidance without altering the optimal solution. All LARMs used in our experiments satisfy this condition.
>
> **Q6) Human verification undermines automation, how often was it needed?**
>
> We see optional human intervention as a feature, not a weakness: natural language makes LARMs a shared, interpretable interface for humans, FMs, and RL agents, enabling transparent and verifiable reward design rather than treating the FM as a black box.
>
> For transparency, Appendix A.4 now reports the exact human interventions. The self-improvement loop handled all DoorKey, Craftium, and all 1,000+ XLand-MiniGrid LARMs (using LLM-as-judge). Human feedback was required for only 3 tasks, to correct a rare logical edge case or improve reward shaping.

---

> ### Author Response · Authors · 2025-11-20
>
> **Q7) How are RM-state language embeddings obtained?**
>
> We used GPT-4o to generate all LARM components for all tasks, with the exception of the 1,000 LARMs for XLand-MiniGrid (these were generated using various open-source FMs of different scales for our ablation study). We acknowledge that we missed specifying this in the paper; we have made it clearer in the revised version.
>
> **Q8) How large are the domain gaps, and can performance be reported as a function of gap magnitude?**
>
> We would appreciate clarification on this question, as we did not explicitly discuss 'domain gaps' in the paper. Could you please elaborate on the specific gaps you are referring to and suggest a concrete metric you would consider using for this analysis?

---

### Official Review · Reviewer_egA8 · 2025-11-06

**Soundness:** 3
**Presentation:** 3
**Contribution:** 2
**Rating:** 6
**Confidence:** 3

**Summary:**

This paper considers the problem of synthesizing reward machines from natural language. The aim then is to use the synthesized reward functions for single and possibly multi-task RL. This works in three parts. First a reward machine language specification is created. This describes the underlying finite state transducer structure of the reward machine, e.g. the states, transitions, alphabet. The second part describes the grounding functions for each input symbol. Finally each state is tagged with a natural language description describing the abstract state of the agent in it's policy. The abstract description is additionally encoded as a text embedding. A key contribution of the work is to propose using this text embedding as the representation of the automata state to the RL-agent. The insight is that this provides two features:

1. If the state is sufficiently descriptive, if provides strong clues as to what the agent should do at that state, e.g., go collect the key for a door.
2. If the state descriptions semantically (or exactly) align, then the policy can re-use experience from other tasks.

Empirical results show improvements against sparse feedback baselines and signs of multi-task generalization.

**Strengths:**

1. Formal representations of tasks such as reward machines offer a powerful mechanism to unambiguously/explicitly provide additional structure and memory to an RL agent. Being able to easily generate such representations is a worth while research program.
1. The exact method used in this paper is to my knowledge novel. While the translation of natural language to an automaton has been explored in other works, this work adds an interesting contribution of a natural language embedding to each state.

**Weaknesses:**

1. The approach for constructing the automaton seems rather ad-hoc and relies somewhat blindly on the LLM's ability to provide a faithful translation. This is in contrast to techniques which seek to reduce the learning of the automaton structure to classic learning algorithms based on membership oracles [1].
2. The language embeddings are the only source of automata structure embedding. it's unclear how reliably the automata's structure is actually relayed. While likely enough for short horizon tasks, the concern is that for complicated automata with many loops, explaining what the state is in natural language is non-trivial. On the one hand, compared to domain agnostic embeddings, e.g., RAD embeddings [2], these provide vital domain specific clues for what the policy should do. I would be curious what would happen when combining such approaches.
3. Perhaps the largest issue is it's unclear how to test for alignment in the work aside from simply optimizing the reward and checking the outcome. The grounding function in particular seems non-trivial to verify without a large amount of human in the loop work.

[1] L*LM: Learning Automata from Demonstrations, Examples, and Natural Language
[2] Compositional automata embeddings for goal-conditioned reinforcement learning

**Questions:**

Have you explored compositional tasks which aren't sequences, but have conditional logic? E.g., If x occurs, then you need to do y. My concern is the state embedding sharing becomes less useful since the local and global automata structure can non-trivially change when adding a task.

---

> ### Author Response · Authors · 2025-11-20
>
> We thank the reviewer for their thoughtful feedback and appreciate their comments that reward machines “offer a powerful mechanism to provide additional structure and memory” and that our method is “novel,” especially in using “a natural language embedding to each state”. Below, we respond to the reviewer’s feedback. **We also encourage the reviewer to read our general comment and the revised manuscript, where all changes are highlighted in blue.**
>
> **Q1) Perhaps the largest issue is it's unclear how to test for alignment in the work aside from simply optimizing the reward and checking the outcome. The grounding function in particular seems non-trivial to verify without a large amount of human in the loop work.**
>
> This is an important point. Verifying alignment is challenging for any method that generates task representations. However, most approaches push verification downstream, after costly data collection and training. For example, [1] requires: (1) collecting an offline expert dataset, (2) learning the DFA, and only then (3) checking correctness indirectly via policy success, an expensive loop.
>
> Our framework moves verification upstream. ARM-FM needs no expert data and outputs an explicit, interpretable LARM before policy learning. This makes verification practical: as shown in Fig. 4, the FM generates both natural language state descriptions and code for labeling functions. Checking alignment reduces to: (1) reading the LARM states, and (2) reviewing small generated predicates (e.g., has_key()).
>
> We now provide more detail in Appendix A.4: this human check was minimal in practice. Moreover, Fig. 11 (left) shows that FMs can themselves evaluate LARM quality, enabling scalable automatic verification.
>
> [1] L*LM: Learning Automata from Demonstrations, Examples, and Natural Language
>
> **Q2) The approach for constructing the automaton seems rather ad-hoc and relies somewhat blindly on the LLM's ability to provide a faithful translation. This is in contrast to techniques which seek to reduce the learning of the automaton structure to classic learning algorithms based on membership oracles [1].**
>
> Approaches like [1] are indeed principled but require an offline expert dataset. Our method targets a different regime: generating reward machines without behavioral examples, using only task descriptions. We now clarify this in the revised Related Work.
>
> We also disagree that our approach relies “blindly” on the LLM. The self-improvement loop (Fig. 3) explicitly tests and refines FM outputs. Strong empirical results across environments show this language-first approach is effective and avoids the cost of collecting demonstrations.
>
> **Q3) The language embeddings are the only source of automata structure embedding. it's unclear how reliably the automata's structure is actually relayed. While likely enough for short horizon tasks, the concern is that for complicated automata with many loops, explaining what the state is in natural language is non-trivial. On the one hand, compared to domain agnostic embeddings, e.g., RAD embeddings [2], these provide vital domain-specific clues for what the policy should do. I would be curious what would happen when combining such approaches.**
>
> We agree that topology-aware embeddings like RAD [2] capture useful structural information. This complements, rather than replaces, our idea: we ground each state in its semantic meaning, which provides actionable domain-specific cues to the policy.
>
> Combining semantic embeddings with structural embeddings is indeed promising. We now highlight this in the future work section and cite [2] in the revised Related Work.
>
> [2] Yalcinkaya et al., “Compositional automata embeddings for goal-conditioned RL.”
>
> **Q4) Have you explored compositional tasks which aren't sequences, but have conditional logic? E.g., If x occurs, then you need to do y. My concern is the state embedding sharing becomes less useful since the local and global automata structure can non-trivially change when adding a task.**
>
> This is a key-point: our method does not rely on sharing a generic, context-free embedding for a sub-task (e.g. a one-hot vector). Instead, the FM would generate disjoint branches in the LARM with distinct states for each logical path. As long as the conditional logic of the task is part of the initial task specification, ARM-FM can handle this directly.
>
> Because our FM generates unique, language-aligned embeddings for these full natural language descriptions, the policy receives different semantic vectors. This is precisely what allows the agent to learn the correct conditional policy and avoids the interference the reviewer is concerned about.

---

### Author Response · Authors · 2025-11-20
**General Comment**

We thank the reviewers for their insightful and constructive feedback. We are encouraged that they (1) found our method novel (reviewers egA8, 3qmz); (2) are convinced of the effectiveness of our method demonstrated by empirical evidence (tffC, 3qmz); and (3) particularly appreciate the proposed mechanism to improve generalization via conditioning on the FM-generated embeddings (4WaP, tffC).

The central contribution of this work is a framework that automates reward design for complex, compositional tasks without requiring expert demonstrations. ARM-FM leverages the reasoning capabilities of FMs to generate interpretable, structured, and verifiable task specifications (LARMs) directly from natural language.

Our key innovation is more than generating the automaton structure, but also generating language-aligned state embeddings. As our results show, this semantic grounding allows the agent to efficiently learn and generalize in complex tasks.

Based on the reviewers' suggestions, we have significantly strengthened the paper in the following ways:

**Clarification of Baselines:** We have explicitly clarified that our primary FM-agent baseline is ReAct, a strong method for in-context reasoning. We have also better contextualized our work with recent related work suggested by the reviewers.

**Transparency on Human-in-the-Loop (Appendix A.4):** We have added a dedicated section and table detailing exactly when and how humans intervened in ARM-FM. We show that the fully automated loop was sufficient for most of the environment used. We also describe the interactive interface that allows humans to seamlessly replace the FM critic for high-level refinement. We view the option for human intervention not as a failure of automation, but as a feature of the ARM-FM framework. By using natural language, LARMs become a common interface that is interpretable by humans, writable by FMs, and learnable by RL agents.

**Path to Real-World Deployment:** We have expanded our discussion on labeling functions, clarifying that while we use code throughout this paper, ARM-FM generally supports any class of labeling functions as long as they evaluate to boolean predicates (e.g., VLM queries or learned classifiers) for real-world application.

**Theoretical Formalization (Appendix A.5):** We have added a theoretical analysis showing that our LARM-augmented objective preserves the optimality of the original tasks. We argue that because our generated LARMs contain no positive reward cycles, the dense rewards act as potential-based shaping that guides exploration without altering the global optimal policy.

We believe that these revisions comprehensively address the points raised by the reviewers. **Please refer to the revised version of the paper, where all changes in the text have been highlighted in blue.**

---

### Meta-Review · Area_Chair_5XiV · 2026-01-07

**Summary:**

ARM-FM introduces a framework that leverages Foundation Models (FMs) to automatically construct **Language-Aligned Reward Machines (LARMs)** from natural language. This method decomposes complex, long-horizon tasks into structured subgoals, generating both the automaton topology and the executable code for labeling functions. A key innovation is conditioning the RL policy on **semantic language embeddings** associated with each RM state, which facilitates zero-shot generalization to unseen tasks.

### Strengths:

- **Methodological Novelty:** Combining the formal structure of Reward Machines with the semantic reasoning of FMs via state-level language embeddings is widely recognized as a novel contribution.

- **Strong Empirical Performance:** The framework significantly outperforms standard RL baselines (e.g., PPO, SAC) and FM-guided agents like ReAct in sparse-reward environments across 2D and 3D domains.

- **Interpretability and Verifiability:** Unlike "black-box" reward models, LARMs provide an explicit, human-readable structure that can be inspected and debugged.

- **Scalability:** The system successfully generated over 1,000 unique LARMs for the XLand-MiniGrid benchmark, demonstrating high-throughput automated design.



### Weaknesses:

- **Reliance on Symbolic State:** The generated labeling functions currently depend on access to simulator-internal variables, which may limit immediate deployment in raw-observation or real-world settings.


- **Human-in-the-Loop Clarification:** While largely automated, some tasks required minimal human intervention, which initially led to questions regarding the "full automation" claim.

- **Structural Representation:** Some reviewers questioned if language embeddings alone sufficiently capture complex, loopy automaton topologies.

### Recommendation for Accept

I recommend **Accept** for ARM-FM. The authors effectively addressed the reviewers' concerns during the rebuttal by adding a transparency section on human intervention (Appendix A.4) and a theoretical analysis of optimality preservation (Appendix A.5). The framework’s ability to bridge high-level reasoning with low-level control through a verifiable, compositional interface represents a significant step forward for instruction-following RL. While the reliance on symbolic predicates is a limitation, the authors' roadmap for plugging in Vision-Language Models (VLMs) as labeling functions provides a credible path for future work.

**Reviewer Concerns:**

### Concerns Addressed by the Rebuttal:

- **Human-in-the-Loop Transparency:** The authors clarified that human intervention was minimal, occurring in only 3 out of over 1,000 tasks, and provided a detailed breakdown in Appendix A.4.

- **Optimality and Theoretical Grounding:** The addition of Proposition 1 formally demonstrates that the FM-generated intermediate rewards guide exploration without altering the global optimal policy.

- **Baseline Comparisons:** The authors clarified that the "LLM Agent" baseline is the strong **ReAct** method and added a comparison table (Table 4) to contextualize ARM-FM against other FM-guided RL frameworks.

- **Generalization Mechanism:** The authors provided evidence that semantic language embeddings allow the agent to generalize across diverse modalities (2D, 3D, and continuous control).

### Outstanding Concerns:

- **Dependency on Privileged State:** While the authors argue the framework is "pluggable," the current experiments still rely on symbolic simulator states (e.g., `has_key`) for labeling functions, leaving its performance on raw sensory data (pixels) for future work.


- **Novelty Debate:** One reviewer remains concerned that using Reward Machines for non-Markovian tasks is well-established, viewing the automation via FMs as an incremental improvement rather than a fundamental shift.

- **Open Source/Reproducibility:** A reviewer noted that the code is not yet publicly available, which may impact the immediate reproducibility and verification of the reported results.

**Reviewer Scores:**

* **Reviewer 3qmz (Score: 8 - Accept):**
This reviewer was already highly positive, noting the work was "insightful and well-executed". Since the authors directly addressed their primary concerns regarding the frequency of human intervention and the role of state memory , this reviewer would have likely **maintained their strong Accept score (8)** or potentially moved toward a **10**, as they found the empirical evaluation "broad and thorough".

* **Reviewer egA8 (Score: 6 - Marginally Above):**
Initially concerned about the "ad-hoc" nature of LLM translation and verification of grounding functions, this reviewer likely would have **increased their score to a 7 or 8**. The rebuttal clarified that the framework moves verification "upstream" by providing interpretable code/descriptions and demonstrated that the self-improvement loop effectively refines the automaton without expert data.

* **Reviewer tffC (Score: 6 - Marginally Above):**
This reviewer explicitly acknowledged that the rebuttal "adequately address[ed] my concerns". While they ultimately kept their score at 6 due to the lack of public code and a perceived "incremental" nature of scaling existing paradigms, a full discussion might have pushed them to a **7**. They already conceded that the "smartly implemented" paradigm is a "good contribution" to scaling RL.

* **Reviewer 4WaP (Score: 4$\to$6 - Marginally Below$\to$ Marginally Above):**
This reviewer’s main critique was limited novelty and "incremental" additions. However, the authors' detailed comparison table and the introduction of Proposition 1 (Optimality Preservation)  provided the technical depth the reviewer felt was missing. Given the clarification that ARM-FM outperforms ReAct and generalizes across 2D/3D modalities, this reviewer likely would have moved to a **6 (Marginally Above)**.

---

### Decision · Program_Chairs · 2026-01-26

Accept (Poster)